# Comparison of the Cisterna Maturation-Progression Model with the Kiss-and-Run Model of Intra-Golgi Transport: Role of Cisternal Pores and Cargo Domains [note 1]

**DOI:** 10.3390/ijms23073590

**Published:** 2022-03-25

**Authors:** Galina V. Beznoussenko, Hee-Seok Kweon, Irina S. Sesorova, Alexander A. Mironov

**Affiliations:** 1The FIRC Institute of Molecular Oncology, Via Adamello 16, 20139 Milan, Italy; galina.beznusenko@ifom.eu; 2Center for Research Equipment, Korea Basic Science Institute, 169-148, Gwahak-ro, Yuseong-gu, Daejeon 34133, Korea; hskweon@kbsi.re.kr; 3Department of Anatomy, Ivanovo State Medical Academy, 153462 Ivanovo, Russia; irina-s3@yandex.ru

**Keywords:** intracellular traffic, Golgi complex, membrane fusion, Golgi dynamics, SNAREs, COPI

## Abstract

The Golgi complex is the central station of the secretory pathway. Knowledge about the mechanisms of intra-Golgi transport is inconsistent. Here, we compared the explanatory power of the cisterna maturation-progression model and the kiss-and-run model. During intra-Golgi transport, conventional cargoes undergo concentration and form cisternal distensions or distinct membrane domains that contain only one membrane cargo. These domains and distension are separated from the rest of the Golgi cisternae by rows of pores. After the arrival of any membrane cargo or a large cargo aggregate at the Golgi complex, the *cis*-Golgi SNAREs become enriched within the membrane of cargo-containing domains and then replaced by the trans-Golgi SNAREs. During the passage of these domains, the number of cisternal pores decreases. Restoration of the cisternal pores is COPI-dependent. Our observations are more in line with the kiss-and-run model.

## 1. Introduction

Proteins and lipids are synthesised in the endoplasmic reticulum (ER) and then transported to the Golgi complex, where they undergo glycosylation and initial sorting, then these cargoes are delivered to their destinations. The precise mechanisms of intra-Golgi transport (IGT) are unknown; however, many models have been proposed to explain this process. Initially, it was believed that the IGT of proteins and lipids follow the compartment progression model, where a cargo-filled compartment also containing Golgi glycolytic enzymes (GGEs) moves to the plasma membrane and fuses with it. However, according to the logic of this model, glycolytic enzymes should be present at the plasma membrane. However, they were not found there, with the exception of mannosidase II (ManII) [1]. 

Studies of the molecular mechanisms of intracellular transport were initiated by the pioneering works of Palade [2,3,4,5]. After this, the vesicular model of transport was proposed [6,7]. Initially, it was believed that vesicles coated with clathrin that were visible near the *trans*-side of the Golgi were transport carriers [8]. Then the model was changed, and vesicles coated with coatomer I (COPI) were assigned as the transporters [9]. However, there are no COPI-coated vesicles near the Golgi complex [10]. 

When we started our studies on the problem of IGT within the framework of the scientific program, according to Lakatos [11], the vesicular paradigm prevailed in this area. However, already at that time, there were observations that mega-cargoes (protein aggregates with sizes > 60 nm not disassembled during IGT) could not be transported by vesicles due to their large size. Thus, the vesicular model has a main prohibiting observation; namely, mega-cargoes cannot be transported across the Golgi stacks by COPI vesicles. However, it was believed that these mega-cargoes were divided into smaller parts during transport, which could be transported by COPI vesicles or by mega-vesicles [12].

At the same time, it was found that GGEs undergo constant recycling. Also, connections were shown between the ER and the Golgi and between the Golgi cisternae [12]. Our analysis of the literature revealed that all of the experimental data also fit into the framework of two new hypotheses, namely, the classical cisterna maturation–progression model (CMPM) and the diffusion model [12]. According to the CMPM, ER-Golgi carriers newly formed from the Golgi cisterna at the *cis*-side of the Golgi stack undergo progression from the proximal to the distal positions due to the constant arrival of new *cis*-cisternae and the constant departure of the cisternae from the *trans*-side of the Golgi stack. The cargo does not leave the cisternae, and the GGEs are recycled into a more proximal cisterna using COPI vesicles. Then this model was modified several times [13,14,15].

In 1998, we demonstrated that procollagen aggregates formed on the *cis*-side of the Golgi stack move through it without undergoing disassembly [16]. This observation completely rejected the vesicular model and was more in line with the maturation model. To explain our data, the vesicular model was modified, and it was assumed that there are special mega-vesicles for mega-cargoes [17]. 

In 2001, using EM tomography, we showed that procollagen aggregates in cisternal distensions are always connected to the Golgi cisternae and thus rejected the mega-vesicular model. All 600 procollagen-containing distensions examined were connected with the cisternae [18]. In addition, our data allowed another interpretation in the form of the carrier maturation model. This model was renamed the asymmetric kiss-and-run model (KARM) [19].

Then we checked the logical consequence of the CMPM, according to which the model will work correctly if the concentrations of the Golgi resident proteins in COPI vesicles are higher than in the cisternae [20]. We found that, contrary to expectations derived from the CMPM, in membranes of COPI vesicles, the concentrations of most GGEs were lower than in the membranes of the cisternae [21].

In 2013, we demonstrated that transporters of nucleotide sugars also have lower concentrations in COPI-dependent vesicles than in the cisternae. Moreover, transporting Golgi stacks had fewer vesicles than non-transporting stacks [22]. Finally, we showed that in *Saccharomyces cerevisiae*, COPI vesicles had lower concentrations of proteins on the Golgi membranes than in the Golgi compartments [23].

At the same time, verification of the diffusion model showed its main consequences and the conclusion that the connections between the Golgi cisternae existed were fulfilled [24]. These connections are filled with albumin [25]. Some lipids can be easily transported along the secretory pathway when the formation of vesicles is inhibited [26,27]. Patterson et al. [28] presented new evidence in favour of the diffusion model of IGT. They showed that the curves of cargo exit from the Golgi area showed negatively exponential kinetics. However, they did not examine the empty Golgi complex. Of interest, lipid diffusion between secretory compartments was observed even after the fixation of cells with formaldehyde [29]. In living cells, spots filled with fluorescent cargoes moved through the prebleached Golgi ribbon, while their intensities decreased gradually [30]. Moreover, dicumarol destabilised Golgi tubules and delayed intra-Golgi transport [31]. In contrast, when the cisternae of the Golgi complex become interconnected, activation of protein kinase A accelerates intra-Golgi transport [32]. Altogether these data provided support for the diffusion model. To solve all of these contradictions, we proposed the KARM, which we had previously used to explain the behaviour of synaptic vesicles [33], for the explanation of IGT [19]. Then the KARM was improved several times [34,35,36,37].

Currently, there are four main models of intra-Golgi transport: (1) the vesicular model; (2) the CMPM; (3) the diffusion model; and (4) the KARM, which exists as symmetric and asymmetric variants. The main principle of KARM is fusion before fission, when fusion results in fission. Fusion/fission might occur at the same site (i.e., symmetric variant) or at different sites (i.e., asymmetric variant). The asymmetric variant had been named as the carrier-maturation model [18,34,35,36,37]. Nowadays, there are more than 20 variants of the 4 main models [15,37,38].

Here, we do not analyse mechanisms of IGT of glycan and mucins because this issue is extremely complicated. For instance, there are N-Linked glycans and O-Linked glycans, which behave as membrane glycosylated proteins. On the other hand, secreted mucins could be secreted (gel-forming [MUC2, MUC5AC, MUC5B, MUC6, MUC19] and non-gel-forming [MUC7, MUC8, MUC9, MUC20]); membrane-bound and transmembrane (MUC1, MUC3A, MUC3B, MUC4, MUC12, MUC13, MUC15, MUC16, MUC17, MUC21, MUC22). However, these proteins could be classified as related to IGT. Glycans, membrane-bound, and transmembrane mucins behave as conventional glycosylated membrane cargo-proteins (i.e., VSVG). The gel-forming glycans correspond to the secretory cargo-aggregates (i.e., procollagen-I), non-gel-forming mucins mimic diffusional secretory cargoes (i.e., albumin [19,25]). Although the intra-Golgi transport is important for the ER-Golgi transport and trafficking system, we examined here only the first issue.

In this paper, we apply approaches for comparison of different models originally reported by Poppers [39], Kuhn [40], and Lakatos [11] to compare the most popular models, namely, the CMPM and the KARM. Here, we show that during IGT, mega-cargoes also undergo concentrating processes. However, due to the size limitations of the diffusion of mega-cargoes along thin intercisternal connections, this is rather difficult. The diffusion model and the KARM will be compared in another report (our unpublished observations). 

## 2. Results

### 2.1. Background

To compare the different transport models to explain the available experimental evidence, we used approaches that were developed by the philosophers of science, namely, Popper [39], Kuhn [40], and Lakatos [11]. According to Popper’s falsification criterion [39], a hypothesis makes sense if it can be defined as false experimentally. This means that the scientist who proposed the hypothesis should describe the prohibiting experiment, one of the possible results of which would refute the proposed hypothesis. This criterion is important to prevent endless adaptations of hypotheses. The authors should describe previously unknown observations that are in favour of their hypothesis. 

On the other hand, the very first version of a hypothesis usually does not fully explain all of the existing and future observations at once. Lakatos [11] established that after the discovery of new evidence that was not predicted by the first variant of a hypothesis, scientists usually adapt this hypothesis to give it the possibility to explain these new observations. Alternatively, after the discovery of prohibiting observations, scientists can reject their own model, as we do in the current paper (see below), because the CMPM was proposed by us [12]. Indeed, initially, we proposed the classical CMPM [13] and then began to adapt it and to perform experiments that would result in prohibiting observations. Indeed, Figure 4B presented by Bonfanti et al. [16] already showed a concentration of PCI-containing distensions at the *trans*-side of the Golgi complex during IGT. However, at that time, we did not pay attention to this. Similar observations were accumulated gradually [41,42,43]. Then, we observed that during IGT, when the transport was synchronised, albumin became enriched at the *trans*-side of the Golgi complex. Albumin was also present inside the tubes that connected the Golgi cisternae [25]. In contrast, proinsulin was not subjected to processes of concentrating [44]. Therefore, again we did not reject the CMPM and assumed that mega- and membrane-cargoes are transported according to the CMPM, whereas albumin moves by diffusion [25].

The strongest prohibitive observation for the CMPM was the enrichment of mega-cargoes at the *trans*-side of the Golgi during IGT, and especially at steady-state (without cargo synchronisation) [15]. This is impossible by definition (Figure 1: I; see Appendix A). Another prohibiting observation for the CMPM was the absence of IGT of mega-cargoes when all of the Golgi compartments are situated at a significant distance from each other such as in S. cerevisiae [23]. Finally, the list of the supporting observations for the CMPM included the necessity of the enrichment of the Golgi-resident proteins in COPI-dependent vesicles [20]. However, we had already established that the nucleotide sugar transporters and most conventional GGEs commonly used as reference proteins for experiments in cell biology were at significantly lower concentrations in the membranes of COPI-dependent vesicles and COPI-coated buds than in the membranes of cisternae [21,22,23]. Although other authors came to the conclusion that the concentrations of the GGEs in COP1 vesicles were higher than in the membranes of cisternae, we carefully analysed their experimental conditions, and illustrations presented in these articles showed that the published data were well explained within the framework of the KARM [38]. A more detailed analysis will be presented below (see Discussion). 

On the other hand, to be efficient, the KARM should be based on the following prerequisites: (1) cargoes can be enriched at the *trans*-side of the Golgi (Figure 1: II); (2) membrane cargoes have to form domains; (3) SNAREs should be concentrated in these cargo domains; (4) these domains have to be separated from the rest of Golgi cisternae by rows of pores or thin tubules, as the sites for subsequent fission (these pores along the cisternal rims are visible in images presented by several groups [18]; see also Discussion, below); however, the functions of the pores remain unknown (Figure 2); and (5) the number of *medial* Golgi cisternae have to be constant, independent of the amount of cargo moving through the Golgi complex [38] (Appendix A; Table 1).

### 2.2. Conventional Mega-Cargoes Are Concentrated at the Trans-Side of the Golgi Complex

Initially, we investigated the main prohibiting surveillance for the CMPM at steady-state (without cargo synchronisation, because the synchronisation itself can affect the parameters of IGT). To this end, we used HepG2 cells (in vitro) and rat liver hepatocytes where IGT was not synchronised, and measured the concentrations of albumin and very-low-density lipoproteins (VLDLs) in the first and last medial cisternae. Figure 3A (quantified in Figure 4A) shows that also at steady-state, the labelling density of albumin in the last medial cisterna was significantly higher than in the first medial cisterna (*p* < 0.05). Of interest, its labelling density was significantly higher near the cisternal rims (*p* < 0.05), both at steady-state and during its synchronous transport. Moreover, the *trans*-most cisterna (TMC) and the trans-Golgi network had the highest concentration of albumin. Figure 3B, C (quantified in Figure 4A) shows that albumin moves mostly along cisternal rims. Then we compared the numerical density of VLDLs in the distensions situated within the first medial cisternae and the last medial cisternae. Figure 3D,E (quantified in Figure 4B) illustrates our conclusion that the numeric density of VLDL in one cisternal distension of the last medial cisterna was significantly higher than that in the first one (*p* < 0.05). Thus, VLDLs are also subject to concentrating at the *trans*-pole of the Golgi complex. 

Furthermore, we measured the numeric density of procollagen (PC)-I–containing distensions in the first and last medial cisternae in human fibroblasts (HFs) at steady-state after stimulation of procollagen synthesis (see Methods). At steady-state, the number of PCI-distensions per 1 µm of length of the last medial Golgi cisternae was significantly higher than that of the first medial cisternae (*p* < 0.05) (Figure 3G,I; quantified in Figure 4C,D).

Next, we tested whether this regularity would also be fulfilled for VSVG. We transfected HeLa and NRK cells with tsVSVG-GFP and incubated the cells at 32 °C. At this temperature, there was no blockage of the release of tsVSVG from the ER. VSVG freely exited the ER and was transported through the Golgi stack. The labelling density of gold labelling for VSVG on the first medial cisterna was significantly lower than on the last medial cisterna (*p* < 0.05) (Figure 3H; quantified in Figure 4E). Unfortunately, we could not examine whether at steady-state, the asialo-glycoprotein receptor (ASGPR) was more concentrated at the *trans*-side of the Golgi, because this protein was also delivered to the Golgi by endocytosis. Altogether, these observations suggested against the CMPM.

Finally, we checked whether, after synchronisation of IGT of PCI and VSVG-GFP, there was similar enrichment of these cargoes at the *trans*-pole of the Golgi. To this end, we stimulated the synthesis of PCI (see Methods) and incubated the cells at 40 °C and in the absence of ascorbic acid in the incubation medium (Table 2). Under these conditions, PC does not come out of the ER. Then we added 100 µM of ascorbic acid and placed the cells at 15 °C for 15 min (mini-wave) or 2 h (maxi-wave). At this time, procollagen was synthesized and accumulated in the output sites from the ER. After incubation at 15 °C, the cells were again placed at 40 °C, and ascorbic acid was removed from the medium. Thus, new portions of procollagen did not come out of the ER anymore. Under these conditions, either small (mini-wave) or large (maxi-wave) amounts of procollagen (which was accumulated in the ER exit sites) was transported through the Golgi complex. We compared the numerical density of the PC distensions per cisterna at 5 min and 10 min after the re-initiation of IGT in the first and last medial cisternae. Within the mini-wave, the numerical density of the distensions was maximal in the last medial cisterna after 10 min. We demonstrated that the number of PC distensions per 1 µm of the length of the Golgi cisternae was significantly higher after synchronisation of the IGT of PCI (*p* < 0.05), although only when the mini-wave synchronisation protocol was applied (Figure 3J; quantified in Figure 4F). or at steady state (Figure 3I; quantified in Figure 4C). With the maxi-wave, the heterogeneity of measurements was so high that we did not obtain any clear conclusions (Figure 3F). Thus, the mean numeric density of PC-containing distensions within the last medial cisterna was significantly higher than in the first medial cisterna (*p* < 0.05). The only interpretation of this observation can be that distensions are concentrated during transport.

Also, VSVG IGT was synchronised (Table 2). Initially, the cells infected with tsVSV were incubated at 40 °C. Under these conditions, VSVG did not leave the ER and accumulated in the ER. Then the cells were placed at 15 °C for 15 min or for 2 h. Under these conditions, small (mini-wave: 15 min at 15 °C) or large (maxi-wave: 2 h at 15 °C) amounts of VSVG accumulated in the ER exit sites (ERES). Then the cells were heated to 40 °C, and cycloheximide (CHM) was added. This resulted in no more VSVG being synthesised, and the VSVG remaining in the ER could not get out of the ER due to the high temperature. Measurement of the labelling density after 5 min and 10 min in the same cisternae showed that with a mini-wave, the highest concentration of gold was in the last medial cisternae after 10 min (Figure 3H; quantified in Figure 4G). With the maxi-wave synchronisation protocol, clear results could not be obtained. Synchronisation of ASGPR was not possible. 

### 2.3. Membrane Dynamics 

The CMPM poses that after the sudden arrival of different amounts of cargo, the number of medial Golgi cisternae can increase depending on these cargo amounts (Table 1). Indeed, according to the CMPM, when several ER-Golgi carriers arrive at the Golgi complex, they fuse and form a new cisterna situated at the *cis*-side of the Golgi stack. However, the amounts of the simultaneously arriving membranes can be different. The sudden arrival of small or large amounts of membranes can have two consequences, namely, the formation of different numbers of new *cis*-cisternae, or the formation of one *cis*-cisterna with a different length. In contrast, according to the KARM, the number of medial cisternae should remain the same, independent of the amounts of membrane arriving, although the *cis*-most cisterna (CMC) would attach to the *cis*-side of the stack of **medial** cisternae, whereas TMC should attach at another pole. Our previous analysis based on the fragmented Golgi complex [24] revealed that such a long *cis*-cisterna was never observed. In contrast, after the arrival of different amounts of membranes at the Golgi complex, the lengths of all of the Golgi cisternae remained the same, but their length increased. Higher amounts of membranes (the maxi-wave) induced higher cisterna augmentation (Figure 5A–D). 

Here, we re-examined this to see if it is valid for the Golgi ribbon. To this end, we accumulated VSVG within the ER by incubating HeLa and NRK cells infected with tsVSV at 40 °C (which prevents the release of VSVG from the ER) for 2 h, then we accumulated VSVG in the area of ERES for 15 or 120 min, and transferred the cells to a water bath at 32 °C for 5 min. At that time, all of the VSVG molecules were already in the Golgi complex, but their exit from the Golgi had not yet been observed [24]. Then we counted the number of only medial cisternae [45] and compared these with a control in which the cells were subjected to exactly the same procedures but were infected with a normal VSV virus. Representative images (Figure 5E–J: quantified in Figure 4H–K) illustrate our observations and measurements. For instance, we showed that the number of **medial** Golgi cisternae was constant both before and after the release in the transport block and did not depend on the amount of VSVG moving through the Golgi complex, whereas the total number increased by 2 (Figure 4I). The arrival of the cargoes resulted in an increase in cisternal length, surface area and volume (Figure 4H,J,K), although the number of medial cisternae remained constant (Figure 4I: black and magenta). Importantly, all of the cisternae in a stack had equal lengths (Figure 4H). This is in agreement with data published by Trucco et al. [24]. In both cases (as predicted, [36]) (i.e., during the mini-wave and maxi-wave; Table 2: items 1, 2), CMC and TMC were attached to the Golgi stacks composed of medial Golgi cisternae. This observation fits well with the KARM.

### 2.4. Membrane Cargoes Form Domains 

Further, we started testing the supporting observation for the KARM because there was only one prohibiting observation for it—the absence of connections between various Golgi compartments in *S. cerevisiae*. Of interest, this situation represents prohibiting observations also for the CMPM and the vesicular model because COPI vesicles are in strings and cannot diffuse from their cisternae [46,47]. Previously we removed this restriction and showed that connections exist [23]. We also demonstrated that when a small amount of VSVG moves through the Golgi complex, it forms distinct fluorescent spots, which do not dissolve after bleaching, whereby after bleaching of half of the Golgi complex, no recovery of the bleached spots was observed (see Movie 1 in [18]). Different methods of immuno-EM have demonstrated that at steady-state and during IGT of VSVG (Figure 6A–D,F–I,K) and ASGRP (Figure 6E,J), distinct domains are formed. The labelling density of VSVG in these clusters was 2.8 ± 0.3 (*p* < 0.05) and 3.1 ± 0.4 (*p* < 0.05)-fold, respectively, higher than in the rest of the cisternae. Also, we measured the distances between the gold particles labelled ManII (10 nm) and VSVG (15 nm) and demonstrated a significantly greater distance between particles of different sizes than between particles of the same size (Figure 6C; quantified in Figure 4L). The concentration of VSVG in Golgi dots was not significantly changed during IGT. When cells transfected with VSVG-GFP were subjected to this multi-wave protocol, and then half of the Golgi area was bleached, no diffusion of VSVG-GFP into the bleached area was observed (Figure 7A shows quatification of the representative images in Figure 8A,B). Thus, VSVG and ASGPR formed domains (distinct domains of VSVG and ASGPR were visible on cryosections; Figure 8G) when they travel through the Golgi complex in small amounts.

Next, we examined whether a membrane cargo preserved distinct domains when its second portions arrived some period of time after the first pulse. To this end, we used RUSH-TNF-α, where at 4 min after addition of biotin, the cells were washed out and again incubated without biotin for 5 min, followed by the second 4 min RUSH release by biotin addition, followed by fixation and visualisation with DAB (Figure 8C). Additionally, we reproduced waves of VSVG using the 40-32 mini-wave protocol with the duration of one pulse for 5 min (see Methods; Figure 8D,E). In both cases, DAB-positive structures were visible within the Golgi complex but near the different Golgi poles. Of interest, VSVG domains co-localised with Ykt6, the R-SNARE (Figure 8F). Similarly, when the multi-wave synchronization protocol was applied for the IGT of PC, aggregates of PCI distensions were found at both sides of the Golgi stack (Figure 8H). Thus, when portions of membrane cargoes moved one after another within the period of the normal time necessary for IGT, they behaved as distinct domains.

### 2.5. Different Membrane Cargoes Form Distinct Domains

We then tested whether distinct membrane cargoes are transported in different domains. To this end, we used HepG2 cells naturally expressing ASGPR and applied the CHM-15-CHM synchronisation protocol (Table 2, item 4). Initially, after the 15 °C temperature block, ASGPR formed spots within ERES, whereas staining for albumin was within the ER (Figure 9A–C). After the release of the block, albumin quickly (within 1–2 min) became enriched within the Golgi zone, whereas ASGPR showed a peripheral spotty pattern (Figure 9D). Finally, ASGPR formed dots within the Golgi zone that was already filled with albumin (Figure 9E). When HepG2 cells were infected with tsVSV and then subjected to the CHM-15-CHM protocol for IGT synchronisation, VSVG and ASGPR formed separate spots within the Golgi area. Overlap between VSVG and ASGPR domains was low (Figure 9F,H–N). Co-localisation between VSVG and ASGPR dots as well as between PC and VSVG dots was low when the CHM-15-CHM protocol was used. At the EM level, these cargoes also formed distinct domains (Figure 8G and Figure 9Q). Conversely, when the maxi-wave protocol was applied, the co-localisation between PC and VSVG dots was significantly higher (Figure 9R; quantified in Figure 4M). Thus, when small amounts of two cargoes are transported through the Golgi complex, these cargoes form distinct and separated domains. 

### 2.6. SNARE Enrichment over Cargo Domains

Further, we tested the next supporting observation for the KARM, namely, the concentration of a set of SNAREs over cargo domains and the consecutive replacement of one set of SNAREs by another during IGT. To this end, we selected for the analysis the following SNAREs: Sec22, Bet1, GS15, GS27, GS28, and Ykt6. Before the release of the transport block, Ykt6 did not co-localise with PCI (Figure 10A), whereas Bet1 co-localised with PCI (Figure 4N). At 3 min after release of the transport block, PCI spots co-localised initially with Bet1 (Figure 10C; quantified in Figure 4N). Co-localisation between PCI and Ykt6 increased (Figure 10B; quantified in Figure 4N). At 7 min after release of the transport block, co-localisation between PCI and Ykt6 was preserved (Figure 10D; quantified in Figure 4N). At 12 min after release of the transport block, co-localisation between PCI and Bet1 decreased (Figure 10F; quantified in Figure 4N), whereas co-localisation between PCI and GS15 increased (Figure 10E; quantified in Figure 4N). Enrichment of Ykt6 over PCI distension was seen at the EM level (Figure 10G–J; quantified in Figure 4N,P). 

Similar dynamics of SNAREs concentration were observed over the VSVG domains when VSVG was synchronised according to the mini-wave or CHM-15-CHM synchronisation protocols (Figure 10K–O; quantified in Figure 4O). This was visible for the central Golgi (Figure 10K,L) and for the fragmented Golgi (Figure 10M–O). However, during the initial phase of IGT, VSVG also co-localised with Sec22 (Figure 4O). Then Bet1 and Sec22 were replaced with GS15 (Figure 10N,O; quantified in Figure 4O). Co-localisation between Ykt6 and VSVG dots was always high (Figure 10L, M; quantified in Figure 4O). Of interest, the labelling density of Ykt6 over Golgi round profiles was low (Figure 4P,Q). GS27 and GS28 were enriched over COPI vesicles and Ykt6, Bet1, and GS15 over cargo domains. Syntaxin 5 was enriched within cisternal rims (Figure 4Q). Thus, during IGT, SNAREs working at the *cis*-side of the Golgi complex that are enriched over cargo domains are replaced by SNAREs, which function at the *trans*-side of the Golgi complex.

### 2.7. Pores Separate Cargo Domains and the Rest of the Golgi Cisternae

Examination of the movies demonstrating how the KARM functions show that during even one passage of mega-cargo, the number of cisternal pores decreased (Figure 2. Appendix A; Table 1). One of the supporting observations necessary for the successful explanation power of the KARM is the necessity to have thin membrane tubules proximal to the cargo-containing domains (‘behind’ the cargo domain; Table 1). This is necessary to facilitate tubule fission and prevent retrograde shift of cargo domains. At the level of the Golgi complex, this demand can be realized by rows of pores. 

Using EM tomography and serial sectioning of hepatocytes at steady-state and of fibroblasts at steady-state or during the synchronous mini-wave protocol IGT, most of the cisternal distensions filled with PC were separated by pores. Figure 11 and Figure 12 show representative images of the rows of pores that separate cisternal distensions filled with VLDLs (Figure 11A–E and Figure 12A) and with PCI (Figure 11F–L and Figure 12B–I; quantified in Figure 7B,C) from the rest of Golgi cisterna. A high proportion of cisternal distensions filled with VLDL or PC (at steady-state and after synchronisation) were clearly separated by several pores (>70%) (Figure 7D). This proportion was similar at both the *cis*-side and the *trans*-side of the Golgi stack (Figure 7C,D). The VSVG (Figure 6E,K) and ASGPR (Figure 6E,J) domains were also separated by pores. Thus, the cargo domains were surrounded by cisternal pores.

The main postulate of KARM is the statement that first, the membranes of the cargo domain and the distal compartment merge, and then the thin tubes that connect the cargo domain and the proximal compartment break. It is, therefore, to be expected, albeit not often, that the distension (filled with procollagen and separated from the rest of the cisterna by a series of pores) will be simultaneously connected by a tube to the distal cisterna. We have shown such images in Golgi ministacks previously (see Figure 3l–n in [24]). Here, we found five similar cases. Figure 11H,J demonstrates this situation. The second consequence of the KARM-dependent events should be the phenomenon that during the shift of the distensions to the next cisterna after breaking the first tubular bridge in a row of pores, the size of the pores located between the cargo domain and the proximal compartment would increase. We also found such images on tomograms (Figure 11A,B,F–K; red arrows). Additionally, we revealed COPI-coated buds on the rims of newly formed large pores situated between the cargo domain and Golgi cisterna (Figure 12E,G,H).

### 2.8. Transport of Different Amounts of Membranes

In most studies, IGT was synchronised. Moreover, to better visualise this transport, large amounts of cargo are usually accumulated before the Golgi complex. Then this bolus is released, and it passes through the Golgi complex as a wave. As such, this might overload the Golgi complex and impair its transport. Thus, the amount of cargo moving across the Golgi complex might be important for understanding the mechanism of IGT. Here, we examined the influence of the amount of cargo on the parameters of IGT. Synchronisation of IGT of PCI and VSVG was performed according to the mini-wave and maxi-wave protocols (Table 2: items 1,2). When a large amount of VSVG or PCI that was accumulated just before the Golgi complex was suddenly allowed to move through the Golgi, after 2 min, these cargoes were detected mostly within the first two medial cisternae (Figure 13A,D; quantified in Figure 7F). In contrast, when a small amount of VSVG or PCI moved, the cargoes filled almost all of the medial cisternae, with the exception of the TMC (Figure 13B,C; quantified in Figure 7E). These distributions of VSVG labelling and PC distensions observed after the mini-wave and maxi-waves protocol were significantly different (*p* < 0.05). 

After the arrival of small amounts of VSVG, the labelling density of GalT decreased 1.9 ± 0.09-fold, whereas, after the arrival of large amounts of cargoes, the labelling density of GalT was decreased by 3.1 ± 0.1-fold. Of interest, when the maxi-wave protocol was used, GGEs and cargoes were often intermixed (Figure 13E). Thus, with an increase in the levels of the cargo during its synchronous delivery to the Golgi complex, the cargo penetrated deeper into the Golgi stack. 

However, not only the level of cargo penetration but also the cargo diffusion mobility might be dependent on the amount of cargo. Indeed, Presley et al. [30] demonstrated that fluorescence recovery after photobleaching (FRAP) of VSVG-GFP localised within the Golgi mass was relatively high. Importantly, they examined large movements of VSVG-GFP through the Golgi complex. In our hands, when small amounts of VSVG-GFP were transported through the Golgi complex, VSVG-GFP–positive spots showed independent ‘hovering’ and did not fuse with each other (see Movie 1 in [18]). Therefore, we tested whether the diffusion mobility of VSVG-GFP depends on the amount of this cargo transported through the Golgi complex. In cells where VSVG-GFP was synchronised according to the maxi-wave protocol, the FRAP of the small bleached area within the Golgi ribbon positive for VSVG-GFP was fast (Figure 13G). Also, the FRAP was fast, especially when the bleaching of the half of the Golgi complex was performed at 3 min to 9 min after the release of the 15 °C temperature block. However, at 9 min to 12 min (when VSVG-GFP reached the very *trans*-side of the Golgi complex), the FRAP decreased (Figure 13H,I; quantified in Figure 7I). Conversely, when a small amount of VSVG-GFP was transported, and then the whole cell less half of its Golgi complex was bleached, the FRAP of VSVG-GFP was low (see Movie 1 in [18]). This is thus in contrast to when large amounts of VSVG-GFP were synchronously transported.

To understand why the FRAP of VSVG-GFP was higher during the maxi-wave, we synchronised VSVG-GFP transport according to the maxi-wave protocol and bleached the Golgi zone at 6 min after the release of the 40 °C temperature block. Then we examined the subsequent arrival of VSVG-containing carriers at the Golgi area. At 4 min after bleaching (when the Golgi was re-filled), the cells were fixed, and the overlap of the Golgi mass before the bleaching and after the second filling of the Golgi was compared by examination at the immunofluorescence level. The first (before bleaching) Golgi mass was coloured in red and the second Golgi mass in green, with aligned images of the cell before bleaching and after its filling, and measurement of the co-localisation between red and green within the Golgi mass. More than 55% of the Golgi area was coloured in yellow, 25% in red, and 15% in green. The overlap was >22% (Figure 13J,K). This suggests that during the second wave, the vast majority of VSVG-GFP had arrived in the area previously occupied by VSVG-GFP during the first wave. Importantly, separate VSVG domains were not evident.

Additionally, we examined serial images obtained using focused ion beam scanning EM technology (Figure 13F; see Methods) in the samples where VSVG-GFP was synchronised according to the maxi-wave protocol, and GFP was labelled with DAB. The contour of DAB-positive structures (traced with red lines, the contours of the empty zones of the same cisternae were contoured with a yellow line, and the other cisternae were contoured in green (Figure 13Q). Figure 13R demonstrates that red structures (VSVG) formed some kind of the ribbon that connected the different Golgi stacks.

Additionally, we carried out three-dimensional reconstruction of the Golgi mini-stacks (after depolymerisation of microtubules according to [48,49] at 6 min after re-initiation of IGT according to the mini-wave and maxi-wave protocols. In cells deprived of microtubules, large amounts of VSVG-GFP travelled across the Golgi complex to form large cargo domains that partially overlapped with the GGEs (Figure 13L,M). Conversely, when only small amounts of VSVG-GFP were transported through the Golgi complex in these cells, the cargoes formed several completely isolated domains without significant overlap between the cargoes and the GGEs (Figure 13N–P).

Next, we tested whether this domain-like organisation of cargoes during IGT might be affected when two cargoes of different types were transported. HFs were infected with tsVSV, stimulated to synthesise PC, and subjected to the maxi-wave or CHM-15-CHM synchronisation protocols. When the maxi-wave synchronisation protocol was used, labelling for VSVG was visible within the PC-I distensions (Figure 9R; see also Figure 1D in Mironov et al. [18]). In contrast, when HFs were examined after their incubation at steady-state and after the CHM-15-CHM synchronisation protocol, the concentration of VSVG in the membranes surrounding PC-I distensions was low (our unpublished observations). Thus, during IGT of PC and VSVG, when moving together according to the mini-wave protocol, their co-localisation was minimal, whereas, after the maxi-wave protocol, VSVG was found in the membrane surrounding PCI-containing cisternal distensions.

### 2.9. COPI Is Important for Restoration of Cisternal Pores

According to the KARM (Appendix A; Figure 2), cisternal pores are consumed during IGT. To confirm this, we measured the numerical density of the pores before the release of the transport block and after the complete passage of PCI through the Golgi complex, according to the mini-wave synchronisation protocol. Before the release of the transport block, the numerical density of the pores over the Golgi cisterna was high (Figure 14A,B; quantified in Figure 7J,K). After incubation of the cells with CHM, the number of pores in the medial cisternae increased, but not after the 40 °C temperature block, as the output of only one type of cargo was blocked. Also, the yellow arrows in Figure 12I demonstrate that at 2 min after the exit block, the medial Golgi cisternae contained significant numbers of pores. After the complete passage of VSVG through the Golgi complex, the density of the pores decreased (Figure 14C,D,F,H,I; quantified in Figure 7J). 

Finally, after an additional resting period of 25 min, their density restored, especially when the synthesis of all cargoes was blocked with CHM (Figure 14E; quantified in Figure 7J). Disappearance of the cisternal pores after the passage of PCI through the Golgi complex of HFs was confirmed using high-pressure freezing of the cells (Figure 14F; quantified in Figure 7J). Disappearance of cisternal pores was also observed in Golgi ministacks (3 h of nocodazole pre-treatment) after the passage of VSVG. There were low numbers of pores in medial cisternae and no CMC and TMC (Figure 14I; see also Figure 7J). Taking into consideration that cisternal pores are consumed during IGT (Figure 2: II), the logical conclusion should be that the number of pores has to be restored for the next transport wave.

We assessed whether the lack of pores might affect IGT. To this end, we applied the ldlF cell-based assay ([22] the sudden inhibition of COPI formation of 52-nm vesicles in ldlF cells by heating to 40 °C for 2 min induces temperature-dependent impairment of the shape of εCOP). Indeed, inhibition of εCOP blocked re-formation of pores; namely, restoration of the numerical density of the pores in the first medial cisterna after IGT and consecutive to its inhibition for 25 min was blocked in the heated ldlF cells (see Methods; Figure 14I,J; quantified in Figure 7J,K). It is also known that addition of AlF_4_ decreased the rate of IGT [15,22,34]. Addition of AlF_4_ blocked restoration of the cisternal pores after the passage of VSVG (Figure 14G; quantified in Figure 7J,K). Similar results were obtained after microinjection of an anti-ßCOP antibody (our unpublished observations). Thus, in the absence of COPI formation of vesicles, the restoration of cisternal pores and the concentration of soluble cargoes was inhibited. 

Finally, we checked whether the delivery of fresh COPI might restore the formation of cisternal pores by the Golgi complex. To deliver fresh COPI into cells where it was impaired by heating, we applied the heterokaryon-based delivery of normal COPI (see Methods). After heterokaryon experiments, we had several cells with two nuclei. These cells can be formed after the fusion of two control CHO cells (with only 15-nm gold particles in endosomes), two ldlF cells (labelled with only 10-nm gold), and one CHO and one ldlF cell. In the last case, the endosomes contained both 15-nm and 10-nm gold particles (our unpublished observations). In the heterokaryon cells where the endosomes contained only 10-nm gold particles (fusion of two ldlF cells), the number of pores after the passage of VSVG according to the mini-wave protocol was the same as that in the CHO cells. In cells where the endosomes contained both 15-nm and 10-nm gold (fusion of one CHO and one ldlF cell), the number of pores in the cisternae was also almost normal. In contrast, in the heterokaryon cells that contained only 10-nm gold (fusion of two ldlF cells), the number of pores in the Golgi cisternae was lower (Figure 7J). Thus, the re-addition of normal COPI into the cells where the function of COPI was impaired restored the formation of the cisterna pores.

Further, we measured the effects of the function of COPI on IGT inhibition using a co-localisation assay. It is known that during IGT, membranes surrounding the cargo domains change the protein compositions of the membrane surrounded by PCI distensions [18,24]. We used the shift in the co-localisation of PCI with GalT as a marker for PCI progression across the Golgi stack. Blockage of COPI turnover with AlF_4_ inhibited PCI progression (Figure 7K).

To determine whether this inhibition would affect the concentration of albumin by the Golgi complex, CHO and ldlF cells were transfected with GFP-albumin, and after 16 h, they were heated at 40 °C for 2 min (which leads to impairment of the COPI formation of vesicles [22]). Then the fluorescent recovery of the bleached Golgi zone was measured. After the heating, the rate of the recovery was slower, and the level of GFP-albumin enrichment at the *trans*-side of the Golgi complex was 2.5-fold slower (our unpublished observations). 

## 3. Discussion 

Several observations can prohibit the CMPM either completely (ability of the Golgi complex to concentrate a cargo during IGT) or partially (different rate of IGT for different cargoes, depletion of GGEs in COPI-dependent vesicles). The necessity for GGE enrichment in COPI vesicles has already been intensively examined. Indeed, according to mathematical modelling, the CMPM works more successfully if the concentration of proteins that reside in the Golgi membranes is higher than in the cisternae [20]. However, these mathematical models did not take into account the real capacity of COPI to form vesicles, which is lower than necessary for the CMPM [22,50]. Also, the concentration of the several resident Golgi proteins in COPI vesicles is lower than in the cisternae [21,22,23,51]. For example, in Figure 2C of Dunlope et al. [50], ManII-HRP was excluded from round profiles. Golgi-resident proteins were also depleted in COPI vesicles visible in *S. cerevisiae* [23]. These data appear to contradict data published by Martinez-Menarguez et al. [52], Gilchrist et al. [51], Rizzo et al. [53], and Pothukuchi et al. [54]. We analysed these data in detail and demonstrated that they could also be interpreted from the point of view of the CMPM ([38], see below). The data from Pothukuchi et al. [54] will also be evaluated below.

This is useful to immediately deliver GGEs to the cargo and to glycosylate this cargo, and then the Golgi enzymes must be removed from the membrane surrounding the cargo domain or containing a membrane cargo inside. We believed that GGEs are removed from Golgi membranes because the thickness of the membrane where the cargo is located is greater, and enzymes do not like a thick membrane, which is preferred by membrane cargoes.

The concentration of a cargo on the *trans*-side of Golgi stacks is the main prohibiting observation for the CMPM. Here, we compared the CMPM and KARM and demonstrated that at steady-state and during synchronous IGT, all of the cargoes examined (including mega-cargos) became enriched at the *trans*-side of the Golgi complex. Many similar images that demonstrate this phenomenon have been published, but explanations were not presented [25,43,55,56,57,58,59,60,61,62,63,64,65]. For instance, in previous papers, Clermont et al. [66] and Rambourg et al. [67] demonstrated that mega-cargoes are enriched at the *trans*-side of the Golgi complex. During IGT, only slight increases in the amylase concentration at the *trans*-pole of the Golgi complex occurs, whereas the concentration of chymotrypsinogen in the last medial cisternae became significantly higher than in the first medial cisterna [60]. Also, Leblond ([42], 1989) presented images in favour of these conclusions. Figures 1A and 2A by Jantti et al. ([43], 1997) demonstrated that the concentration of the virions of the Semliki forest virus (a mega-cargo) in the *trans*-cisternae was higher than in the *cis*-cisterna.

Thus, during IGT, not only soluble cargoes, but also mega-cargoes, became enriched at the *trans*-side of the Golgi complex. Here, we demonstrated that also all conventional cargoes examined in the last medial cisterna have higher concentrations than in the first medial cisterna, both at steady-state and after different modes of cargo synchronisation. Although these observations are prohibitive for the CMPM [38,68,69] almost all conventional mega-cargoes are enriched at the *trans*-side of the Golgi complex.

We have shown that not only mega-cargoes, but also membrane cargoes, form distinct domains at steady-state and during IGT, but only when small amounts of membrane cargo were transported. VSVG-GFP–positive spots move through the Golgi complex as non-dividing and non-dismantling spots [18]. Different membrane cargoes formed distinct domains (current report; and Figure 4j of Boncompain et al. [70]). Also, it is known that different cargoes exit the ER in different carriers [71]. The domain organisation of cargoes during IGT was less evident when large amounts of cargoes moved. Under these conditions, when a cargo is released suddenly, this cargo penetrates deeper into the Golgi stack, whereas when a small amount of cargo is moved, the level of this penetration is significantly lower. Our data explain why Patterson et al. [28] found penetration of VSVG through the whole stack up to the MTC. Analysis of published papers revealed that Patterson et al. [28], Griffiths et al. [72], and Bergmann [73] examined the situation when large amounts of VSVG moved through the Golgi complex. Additionally, Trucco et al. [24] demonstrated that the volume of the Golgi increased 2.3-fold when the mini-wave synchronisation protocol was used. At steady-state, the amount of cargo that moves across the Golgi complex was 2.8-fold lower than if the mini-wave synchronisation protocol was used. In rat hepatocytes, 47% of the Golgi complex is composed of cargoes [63]. Thus, even the mini-wave synchronisation protocol induced overloading of the Golgi complex.

When the amount of a membrane cargo was large, the cargoes formed a quasi-separate cargo ribbon in a parallel manner along the ribbon filled with the GGEs and were in close contact with the GGEs. In the experiments by Boncompain et al. ([70], their Figure 4d–g), TNF-α also does not enter into the ManII zone. When the amount of VSVG-GFP passing through the Golgi complex was high, spots underwent dissolution (a spot suddenly disappeared [30]), as it been described by Polishchuk et al. [74] for the plasmalemma. The situation is similar to that observed at the level of the plasmalemma when post-Golgi carriers fuse with the plasma membrane [75,76].

On the other hand, the CMPM poses that ER-Golgi carriers arrive at the Golgi complex and form the new *cis*-cisterna. However, when a large amount of cargo arrives, there should be either formation of several new cisternae or formation of a very long *cis*-cisterna. In contrast, the KARM poses that after the sudden and massive arrival of cargo at the Golgi complex, the total number of cisternae in the Golgi stacks would increase by 2; namely, the CMC and TMC will attach to the stack of Golgi medial cisternae, whereas the number of medial cisternae would remain constant, although their length increases because the CMC and TMC are attached to the Golgi complex when IGT is in an active stage [77].

According to the KARM [38], enriched SNAREs within cargo domains increases the efficiency of IGT. Indeed, membrin (GS27) and GOS28 (GS28) were enriched in Golgi COPI-dependent vesicles, whereas Bet1, GS15, Ykt6, and syntaxin 5 were depleted. Ykt6, Bet1, and GS15 were enriched over cargo domains, whereas syntaxin 5 was on conventional cisternal rims. During IGT, concentration of *cis*-SNAREs over the cargo domains increased. Then, the *cis*-SNAREs that had been enriched in the cargo domains were replaced by the *trans*-SNAREs. SNAREs were enriched on the cargo domains situated in the nearest proximal Golgi cisterna. On the other hand, GGEs [21] and syntaxin 5 were enriched at the corresponding cisternal rims [21]. This potentiates the progression and glycosylation of cargoes. Importantly, most of the distensions of Golgi cisternae containing cargoes were separated from the rest of the Golgi cisternae by pores. This fits well to the KARM. When the exit of cargo was blocked with CHM, the number of pores increased maximally [63]. 

In the literature, there are many similar phenomena in images presented that have remained without explanations [38,55,56,57,77,78,79,80]. Our data suggest that pores are consumed during IGT and that the formation of pores is COPI-dependent. Previously, we demonstrated that COPI is not important for VSVG and albumin diffusion through Golgi stacks [22]. Also, Taylor et al. ([63], see Figure 1 in this paper) demonstrated that after blockage of cargo synthesis, the numerical density of pores in cisternae increased. Previously, we demonstrated that the concentration of GGEs was higher near the cisterna rims with marginal pores [21], whereas near the rim not containing marginal pores, the concentration of GGEs was lower. This indicates that the zone without such pores contained cargo domains. For tubule fission, it is necessary to have fission machinery, like endophilin and BARS [81], ARFGAP [82], and a different organisation of the lipids (i.e., membranes of COPI buds are thinner than in a cisterna [83]). An important observation is the visualisation of COPI-coated buds on the edges of the large pore that are presumably formed after the fission of a tubule separating the pores in such a row (Figure 14E,G,I). This observation suggests that COPI also participates in the fission of the tubules that separate the pores in the row of pores. 

It has been proposed that COPI vesicles form tubules and thus might be responsible for restoration of cisternal pores; namely, pores might be the result of the retrograde fusion of COPI vesicles [84,85]. Here, we tested this proposal experimentally. First, blockage of GTP hydrolysis and elimination of the formation of 52-nm vesicles by COPI slows IGT and blocks restoration of the pore pool. Hirschberg et al. [75] and Pepperkok et al. [86] also demonstrated that in the presence of AlF_4_ and GTPγS, IGT was slower. Importantly, when all of the SNAREs were inhibited, IGT was also blocked (our unpublished observations). COPI is involved in the fission of COPI vesicles [87]. Previously we demonstrated that the number of 52-nm Golgi vesicles is inversely correlated with the number of inter-cisternal connections [22]. It might be assumed that after their COPI-dependent formation and consecutive fusion with cisternae, COPI vesicles generate the excess of membrane curvature that leads to the formation of tubules and then to the backward fusion of the tip with the cisterna [19]. This mechanism might be responsible for the restoration of cisternal pores. Of interest, tubulo-vesiculation of the Golgi accelerates the IGT of GGEs [88]. COPI has several additional functions. The COPI machinery is important for the separation of ER-Golgi carrier membranes from the ER [71,89], for the exit of GGEs from the ER, and for their concentration. COPI extracts curvature from the Golgi cisternae [90,91]. Then, it could generate tubules and pores. The mechanism possibly involving COPI in this process is explained in [91]. COPI vesicles extract Qb SNAREs from the Golgi cisternae. COPI also forms 50-nm vesicles at the level of the ERES [22,92]. COPI vesicles might be the mechanism responsible for Golgi uncoating. Indeed, when COPI formation of 52-nm vesicles is affected, the Golgi uncoating is slower [93].

In general, the main prohibiting observation for the KARM was the existence of the Golgi complex as separate compartments that are not attached to each other and cannot fuse with each other (e.g., in *S. cerevisiae*). Under these conditions, KARM cannot explain how ITG occurs. However, recently, we showed that the Golgi compartments in yeast could fuse with each other and form a common compartment [23,94]. This makes this Golgi compatible with the KARM [18]. Also, Casler et al. [95] and Kurokawa et al. [94] demonstrated experiments that appeared to be only in favour of only the CMPM, at least in yeast. However, we have shown [18] that the KARM does not deny the process of domain maturation and is compatible with their data. Importantly, neither the KARM nor the diffusion model denies the phenomenon of cisterna maturation.

A similar situation was reproduced in mammalian cells; namely, Golgi stacks were disassembled, and different single Golgi cisternae were attached to mitochondria [96]. Under these conditions (when individual Golgi cisternae are separated and “land-locked” between mitochondria), cargo processing and transport continue [96]. Small cargoes pass through synthetically ‘glued’ Golgi, supporting the stable compartments model in mammalian cells, and suggesting that cisterna progression is not required for anterograde IGT of cargoes [47]. However, careful analysis of the images presented revealed that under these conditions, mitochondria formed aggregates where Golgi cisternae were situated close to each other; all of these mitochondria were grouped in the centre (according to Figure 3 in Dunlop et al. [96]), and the tubules of the cisternae could easily make contact with each other. This makes possible their contact and membrane fusion, similar to that seen in *S. cerevisiae* [23,94]. Tubulation of the Golgi complex led to enhanced trafficking of the VSVG, the cell adhesion protein integrin, and the lysosomal enzyme cathepsin D [97], as well as CD8 [98] and Golgi enzymes [88]. However, the role of Ca^2+^-dependent fusion is high [99,100,101,102], which is once more against permanent membrane continuities and is in favour of the KARM. We have already demonstrated that ManII-positive Golgi compartments form ribbons [19,99]. Here, we described the formation of the cargo ribbon when the Golgi complex was overloaded (Figure 15: I). Figure 15: II shows IGT when there is not overloading.

In summary, when the formation of COPI vesicles is inhibited, the CMPM cannot explain why monomeric VSVG reaches the *trans*-side of the Golgi stack faster than procollagen-I [25]. The CMPM cannot explain the observation that demonstrates that when cargo exit from the ER was blocked in *S. cerevisiae*, the Golgi complex disappeared [103,104], whereas the KARM can [38]. The CMPM cannot explain why COPI vesicles have a lower concentration of resident proteins than in the cisternae, which are located in Golgi cisternae, but the KARM can. The CMPM cannot explain why the number of medial cisternae is constant when transporting cargo through the Golgi complex and does not depend on the availability of cargo and its different quantities. The CMPM cannot explain why connections are formed between cisternae after cargo arrival and why the length of all of the Golgi cisternae is the same during transport [24]. The CMPM cannot explain why a domain organisation of membrane cargoes and mega-cargoes is needed, on which the concentration of SNAREs is increased and which are separated by a number of pores from the rest of the cisterna, but the KARM can. The CMPM cannot explain why COPI vesicles are needed if there is a reduced concentration of Golgi resident proteins in them, and the KARM can; namely, COPI is needed to remove the curvature that is concentrated in COPI vesicles [90,105,106] and to block the fusion of cisternae, as COPI vesicles are depleted of GS27 and GS28 from the membrane of the cisternae, which blocks the formation of the functional SNARE complex [22]. Finally, the KARM fits well to the observation that suggests that expression of the syntaxin-5 binding deficient mutant of Golgin45 induces fusion of the neighbouring Golgi cisternae [107]. 

On the other hand, the CMPM can explain why the Golgi disappears in yeast when cargo delivery is blocked there, but the KARM can also [38]. On the other hand, if the disappearance of the Golgi in yeast (*S. cerevisiae*) according to the CMPM is possible, then why does it not disappear in other types of cells [63]. The KARM easily explains this contradiction [38]. Thus, the KARM is a more powerful model of IGT than the CMPM. 

### How to Reconcile the Hypothesis with Previous Data and Models 

In 2013, Rizzo et al. [53] tried to revive the CMPM. We had already analysed their data and shown that their results can also be interpreted in favour of the KARM [38] (see also details in the Supplementary Material of [38]). For instance, in cryosections, round profiles coated with COPI mostly represent cross-sections of Golgi tubules, most of which are coated with COPI [108]. However, they did not use GS27 and GS28 as markers of real COPI-dependent vesicles [24]. They also did not check the fate of ManI polymers at 20 min and more after its polymerisation. However, insoluble polymers shown to be formed within the Golgi complex were shifted to the endosomal compartment [109], and this phenomenon is compatible with the asymmetric variant of the KARM. We have also provided an alternative explanation for the data (that is suitable for the KARM) obtained in *S. cerevisiae,* and that appeared to be in favour of the CMPM [38].

Recently, Pothukuchi et al. [54] concluded that binding of GRASP55 to molecules of GGEs synthesising lipids prevents enzyme entry into COPI vesicles and thereby blocks retrograde transport of these enzymes in COPI vesicles. For this analysis, they considered round profiles with diameters of 50–80 nm in the peri-Golgi area as COPI-derived vesicles. However, they did not perform three-dimensional EM analysis to show that these round profiles were really vesicles, and they did not label these vesicles for GOS28 or GS27 (membrin), the markers of COPI-dependent vesicles [22,24]. Moreover, all of the profiles in the images presented that are labelled as COPI vesicles are oval-shaped. This is in significant contradiction with the data about the very stable diameter of real COPI vesicles (52 nm, according to Marsh et al. [10]). Therefore, the round and ovoid profiles which they considered as vesicles, might be cross-sections of tubules. This can be attributed to the compression of the sections. However, in the same images, there are absolutely round profiles with diameters of 52 nm (see Figures 3 and 5 of Pothukuchi et al. [54]), which makes this assumption of the sections being compressed unjustified. We believe that their calculations were based on the use of both COPI vesicles and cross-sections of the Golgi tubules. This proposal is reasonable because the number of Golgi tubules increases when the GRASP55 and GRASP65 proteins are removed from cells [110]. This increase might mimic the accumulation of COPI vesicles. 

Other published data on higher concentrations of GGEs in COPI vesicles after isolation of these vesicles in vitro [51] might be explained by artefacts of biochemical preparation [38]. A slight increase in the GGE concentrations in cells [52] is associated with the incorrect sampling of COPI vesicles [38]. Indeed, the authors described that they considered circular profiles covered with COPI located near the Golgi cisternae as COP1 vesicles. The labelling density in these was less than the labelling density of ManII in the second cisterna, where most of the ManII is localised [52]. Moreover, all of the COPI vesicles in the Golgi region were not coated with COPI [10]. Therefore, although we do not specifically examine this issue here, we measured the concentrations of ManI, ManII, GalT, and STF, and again, we showed that these are less than in the Golgi cisternae (our unpublished observations). Therefore, we can conclude that there are no convincing data that exhaustively demonstrate that the concentrations of GGEs in COPI vesicles are higher than in the Golgi cisternae. Of interest, Ladinsky et al. [111] also noted pores between membrane domains (which presumably contained a cargo) and the rest of the Golgi cisterna. Thus, in COPI vesicles, the GGE density is lower than in the cisternae themselves.

Thus, our analysis has revealed that the conclusion about the involvement of COPI vesicles in retrograde transport of GGEs is not justified, and as such, these data do not contradict the KARM (Figure 15: II).

## 4. Materials and Methods

### 4.1. Reagents and Treatments

The sources of reagents, cells, and constructs are indicated in Table 2. Reagents were used according to the manufacturer’s instructions. All antibodies were subjected to all necessary routine control testing, and the numbers of background and presumed positive labelling obtained after these tests were used for the calculations and statistical analysis.

**Table 2 ijms-23-03590-t002:** Reagents and algorithms.

Reagent or Resource	Source	Identifier
**Antibodies**		
Rabbit polyclonal antibody against GFP	Abcam (Cambridge, UK)	ab6556
Rabbit polyclonal antibody against syntaxin 5	Abcam (Cambridge, UK)	ab211417
Rabbit polyclonal antibody against syntaxin 5	Santa Cruz Biotechnology (USA)	Sc365124
Rabbit polyclonal antibody against Bet1	Abcam (Cambridge, UK)	ab42488
Rabbit polyclonal antibody against GM130	Abcam (Cambridge, UK)	ab31561
Monoclonal antibody against VSVG (P5D4)	Abcam(Cambridge, UK)	ab50549
Rabbit polyclonal antibody against mannosidase II	Abcam(Cambridge, UK)	ab12277
Rabbit polyclonal against asialoglycoprotein receptor	Abcam(Cambridge, UK)	ab42488
Rabbit polyclonal antibody against albumin	Abcam (Cambridge, UK)	ab42488
Rabbit polyclonal antibody against Ykt6	Abcam(Cambridge, UK)	ab236583
Rabbit polyclonal anti-human antibody against β-1,3-galactosyltransferase 5	MyBioSource Inc. (San Diego CA. USA)	MBS2026257
PCI polyclonal against the α1 chain C-terminal domain of PCI	L. W. Fisher (National Institutes of Health)	N/A
Affinity-purified polyclonal sheep antibody against human pro-collagen I α1	R&D Systems, Inc. (NY, USA)	AF6220
Rabbit polyclonal antibody against ßCOP	Dr J. Lippincott-Schwartz (Howard Hughes Medical Institute, Ashburn, VA, USA)	N/A
Rabbit polyclonal antibody against GS15	MyBioSource, Inc. (San Diego CA. USA)	MBS2525056)
Rabbit polyclonal antibody against membrin	Biorbyt Ltd. (NY; USA)	orb185573
Rabbit polyclonal antibody against GOSR1(GS28)	Biorbyt Ltd. (NY; USA)	orb326297
Rabbit polyclonal antibody against TGN46	S Ponnambalam (Leeds University, UK)	N/A
GFP-albumin	Beznoussenko et al. [25]	N/A
Fab fragments of polyclonal antibodies conjugated with HRP	Jackson ImmunoResearch (West Grove, PA, USA)	Not produced anymore
Fab fragments of the polyclonal antibodies against IgGs	Jackson ImmunoResearch (West Grove, PA, USA)	Not produced anymore
Mouse IgG rabbit polyclonal antibody	ThermoFisher Scientific (USA; https://www.thermofisher.com, accessed on 15 March 2022)	31188
Goat anti-rabbit, anti-mouse, and anti-sheep IgG antibodies conjugated with Alexa 488, Alexa 546	ThermoFisher Scientific (USA; https://www.thermofisher.com, accessed on 15 March 2022)	Many
Goat anti-rabbit Fab’ fragments of IgG conjugated with Nanogold	Nanoprobes (Yaphank, NY, USA)	#2004-1 mL
Protein A conjugated with gold particles of different sizes	Utrecht Universit. Utrecht; The Neverland)	Direct order
**Chemicals, peptides, and recombinant proteins**		
Gold enhancer	Nanoprobes (Yaphank, NY, USA)	
Reagents for electron microscopy	EMS or Sigma (Merck)	Many
FUGENE6 transfection reagent	Roche (Monza, Italy).	
**Recombinant DNA**		
Plasmid containing α1-chain of PC tagged with GFP	Beznoussenko et al. [25]; Patterson et al. [28]	N/A
Plasmid with GFP-Alb	Beznoussenko et al. [25]	N/A
Plasmid with PCIII-GFP	Perinetti et al. [112]	N/A
**Software and algorithms**		
Image J	NIH, Bethesda	N/A
IMOD 4.0.11 package	The Boulder Laboratory for 3-D Electron Microscopy of Cells. University of Colorado Boulder	http://bio3d.colorado.edu/imod/accessed on 20 January 2020
Prism: Version 9.4.2 package	GraphPad (NY; USA)	https://www.graphpad.com/scientific-software/prism/ accessed on 20 January 2020

### 4.2. Cells

Human fibroblasts, HeLa cells, NRK cells, HepG2 cells (from the same sources as described previously [18,21,22,23,24,25]; see Table 3) were grown at 37 °C in a 5% CO_2_-humidified atmosphere (CO_2_ incubator). HFs were from Dr. M. De Luca (Istituto Dermatopatico dell’Immacolata, Rome, Italy). The ldlF mutants of CHO cells (from Dr. M. Krieger, Massachusetts Institute of Technology, Cambridge, MA, USA) and wild-type CHO cells were cultured according to Guo et al. [113]. Cell cultures were maintained in DMEM supplemented with 10% (*v*/*v*) foetal (or adult) bovine serum and 50 mg/mL penicillin, and streptomycin (Biological Industries; now Sartorius; USA; https://www.sartorius.com/en/research-areas, accessed on 15 March 2022).

### 4.3. Animals

We worked with animals exactly as described earlier [41] using the same permissions. Briefly, 10 6-month-old male Sprague-Dawley rats and 25 6-month-old male Wistar rats (from Cardiological Centre in Moscow, Russia) were used (considering minimal use of animals to validate results). The animal facilities of the Ivanovo State Medical Academy housed animals in plastic sawdust-covered cages on a 12 h dark/light cycle, under standard conditions (room temperature, standard rat pelleted food, and water *ad libitum*). In all experiments, rats were matched for age (6 months) and sex (males). Rats were anaesthetised with a combination of Zoletil (active substances: zolazepam hydrochloride, thiamine hydrochloride, in equal proportions) and 2% Rometar, as described previously [41]. 

The experiments were approved by the Ethics Committee of Ivanovo State Medical Academy (№1 from 5/XII, 2018), in compliance with Order No. 755 of the Ministry of Health of the USSR of 12 August 1977. “On measures to further improve the organisational forms of work using experimental animals” and a letter from the Ministry of Agriculture dated 22.02.05 No. 13-03-2/358, “On modern alternatives to the use of animals in the educational process”. All experiments on live animals were carried out in Russia; the samples were irreversibly fixed with glutaraldehyde, embedded in Epon, and then transported to Italy, where they were examined. 

### 4.4. Constructs

Dr. J. Lippincott-Schwarts (Howard Hughes Medical Institute, Ashburn, VA, USA) kindly sent the cDNA of GalT-CFP, GalT-GFP, and VSVG-YFP. The PC-I-GFP and PC-III-GFP chimaera proteins were characterised previously [28,112]. Cells were transfected with the constructs exactly as described previously [22].

### 4.5. Experiments with Cells

The infection of cells with the ts045 strain of vesicular stomatitis virus (tsVSV) and their transfection with different fusion proteins, transfection of cells, and stimulation of PC synthesis in HFs, was all performed as described previously, in the presence of 100 µM ascorbic acid [114]. Isolation and fixation of chicken fibroblasts were described by Bonfanti et al. [16]. Synchronisation of IGT for cargoes was performed as described previously [18,24]. All protocols of the cargo synchronisation are explained in Table 3. The CHM-15-CHM synchronisation protocol was composed of initial treatment of cells with 100 µg/mL CHM at 37 °C to block synthesis of all cargoes in the ER. After 1 h, the CHM was eliminated, and the cells were placed at 15 °C for 15 min or 2 h. Next, the cells were placed at 37 °C (or 32 °C for VSVG and ldlF cells), and 100 µg/mL CHM was added again. Under these conditions, only cargo that was concentrated within the ERES can travel across the Golgi complex. To study the role of COPI in reformation of cisternal pores, we infected CHO and ldlF CHO cells (a CHO derived cell line with a single point mutation in ε-COP, which results in temperature-sensitivity) with tsVSV and subjected to the CHM-15-CHM synchronisation protocol. Just before release of the transport block, both types of cells were heated to 40 °C for 2 min (which leads to detachment of ε-COP and impaired formation of vesicles by COPI [22]). When the transport block was released (CHM-15-CHM protocol; Table 2), the cells were shifted to 32 °C, and the numerical density of cisternal pores was estimated during the time course indicated in the Figures. Additionally, to study the role of COPI in concentration of soluble cargo during synchronous IGT, we transfected CHO and ldlF CHO cells with GFP-albumin and applied the CHM-15-CHM synchronisation protocol, and then examined the FRAP after bleaching of the Golgi area [25]. In HFs, steady-state was realized after stimulation of synthesis of procollagen and addition of 50 µM ascorbic acid for 6 h. Two-wave synchronisation IGT was realized in the following way. VSVG was accumulated within the ER during a 2 h incubation of cells at 40 °C. Then, cells were shifted to 32 °C for 5 min, and then back to 40 °C for 3 min, again to 32 °C for 5 min, and finally once again back to 40 °C. To mimic steady-state transport waves, we used the following consequence of temperature shifts: 3 h at 40 °C, 5 min at 32 °C; 12 min at 40 °C; 5 min at 32 °C; 5 min at 40 °C, and then fixation. CHM was used at 100 µg/mL. Fragmentation of the Golgi complex was achieved as described using 10 µg/mL (33 µM) nocodazole; namely, non-permeabilised cells were treated with 33 µM nocodazole for 3 h to produce ministacks [24,49].

### 4.6. Heterokaryon Assay

When cells attached to glass were treated with polyethene glycol (PEG), cells with two or more nuclei were observed relatively rarely. In contrast, when cells in pellets were treated with PEG, the frequency of this phenotype was higher. Therefore, we used mostly the protocol based on the pellet treatment. The heterokaryon experiments were organised as follows. Initially, ldlF cells underwent 2 min of heating to 40 °C to destroy coatomer, according to Fusella et al. [22]. Then ldlF cells were incubated with wheat germ agglutinin (WGA) conjugated with 10-nm gold for 15 min to load the endosomes. Control CHO cells were incubated with WGA conjugated with 15-nm gold for 15 min. Further, both control CHO and heated ldlF cells were detached, mixed with each other (1:1), and treated with 50% PEG; they were additionally incubated under normal conditions. To induce cell fusion but decrease the toxicity of PEG1500 (Sigma-Aldrich; Milan; Italy; pH 7.8) and to increase the frequency of cell fusion, multiple treatments with 50% PEG was used, with washout. Also, to reduce toxicity, 10% dimethylsulphoxide was added. In this case, the concentration of PEG was reduced to 35%. Pre-treatment of cells with WGA increased the number of cells with two or more nuclei. Using this approach, we increased the frequency of cell fusion by 5-fold. To decrease the toxicity, PEG with the molecular mass of 1500–2000 was used to induce cell fusion, with the time of contact with the cells limited to 1 min. The pellets were thus treated with 50% PEG for 1 min, then placed in 25% PEG and then in medium. The second round of the cell treatments with PEG was the same, with a third-round also applied; the cells were then used for EM.

### 4.7. Fluorescence Microscopy

Light microscopy and EM were performed as described previously [25]. Briefly, the cells were fixed at the desired times using 4% paraformaldehyde and then processed for immunofluorescence, as described previously [25]. Fluorescence microscopy analysis included time-lapse analysis and measurement of FRAP (LSM510 Meta confocal system; Carl Zeiss, Germany) with a 40× objective lens using the original manufacturer software, as described previously [24]. The level of co-localisation of two markers was measured as described previously [25]. Brightness and contrast of the fluorescent images were adjusted (Photoshop CS5.1), and all of the live confocal images and the wide-field images were Gaussian filtered (rotationally symmetric low-pass filter). Red and green pseudo colouring was used, as indicated. Only dots with intensities of >2-fold background were counted. 

### 4.8. Electron Microscopy

Conventional EM, three-dimensional reconstructions, correlative video–light EM, immuno-peroxidase and nanogold EM labelling, serial ultra-thin cryo-sectioning, counting of labelling density, and EM tomography were all carried out as previously described [115,116]. Manipulations were carried out at room temperature (~21 °C) unless otherwise stated. After washing in distilled water, the cells were postfixed in 1% OsO_4_ that contained 15 mg/mL K_4_[Fe (CN)_6_] in 0.1 M sodium cacodylate buffer, at room temperature for 1 h, then washed in distilled water, dehydrated through a graded series of ethanol, infiltrated with Epon 812 resin for 1 h, and polymerised at 60 °C for 24 h. High-pressure freezing was performed using sapphire disks, with all procedures as described previously [71]. Ultrathin sections were collected on pioloform-coated grids and viewed under EM (Tecnai 20; FEI/ThermoFisher Scientific, Eindhoven, The Netherlands). Correlative video–light EM analysis was performed as described previously [24,115,116].

### 4.9. Pre-Embedding Immuno-Electron Microscopy 

Nanogold enhancement of nano-gold particles was performed as described previously [117]. Briefly, the cells were grown on glass coverslips to semi-confluence. The cells were fixed with 2% formaldehyde and 0.05% glutaraldehyde in Na-cacodylate buffer (pH 7.2) at room temperature for 10 min, washed for 3 min to 5 min with phosphate-buffered saline (PBS) including glycine (20 mM sodium phosphate, pH 7.4, 150 mM NaCl, 50 mM glycine), to remove aldehydes, for 3 min to 5 min with PBS–bovine serum albumin–Tween (PBS containing 1% bovine serum albumin, 0.05% Tween 20), and then for 3 min to 5 min with 5 mM sodium phosphate, pH 5.5, 100 mM NaCl (as solution E from the gold enhancement kit; GoldEnhance-EM 2113; Nanoprobes, Inc. (Yaphank, NY, USA). The cells were blocked with 5% bovine serum albumin, 5% normal goat serum, and 0.1% cold-water fish skin gelatine in the permeabilisation buffer with 0.005% saponin. The cells were labelled with antibodies against the antigen of interest for 1 h at room temperature. The cells were washed in permeabilisation buffer containing 0.01% saponin and 0.1% bovine serum albumin and incubated in goat anti-mouse or anti-rabbit Fab fragments conjugated to 1.4-nm nanogold particles. The cells were then washed with permeabilisation buffer and fixed with 1.5% glutaraldehyde in permeabilisation buffer for 10 min. After washing, the gold labelling was intensified using gold enhancement kits (Nanoprobes; Inc. Yaphank, NY, USA). Then, immunoEM labelling was performed, with the final stage of Fab fragments conjugated with nanogold. For gold enhancement, the cells were placed in a mixture of the manufacturer Solutions A and B at a 2:1 ratio (80 µL A, 40 µL B). After 5 min, 200 µL solution E with 20% gum Arabic was added, and then 80 µL of solution C. After 7 min to 15 min of development, the samples were transferred to 2% sodium thiosulphate to terminate the enhancement. The samples were then washed for 3 min to 5 min with buffer E. This protocol resulted in slower development, reduced background, and improved uniformity of enhanced particle size, especially when the enhancement was carried out at 4 °C rather than room temperature. Gold-enhanced, chemically fixed samples were incubated in 1% OsO_4_ in 0.1 M sodium phosphate, pH 6.1, for 60 min, and then rinsed with distilled H_2_O prior to 1 h staining *en-bloc* with 2% uranyl acetate, dehydration and embedding in Epon. Silver enhancement was performed exactly as it was described [118].

### 4.10. Cryosections

Cryosection-based immuno-EM analysis was performed as described previously [23]. Cryosections of cells prepared according to the Tokuyasu method were immuno-labelled with antibodies against the proteins indicated in the text and in Table 2. For immuno-fluorescence, the dilution of antibodies was used according to the manufacturer’s instructions. The EM dilution was 10-fold lower. Thawed cryo-sections with 45 nm to 50 nm thickness were labelled with an antibody (usually with a dilution of 1:50) against the antigens of interest (Table 2) and then stained with protein A conjugated with 10-nm gold particles. After contrasting with uranyl acetate, these were examined under EM (Tecnai 20; FEI/ThermoFisher Scientific). The blocking solution contained 0.1% cold-water fish skin gelatine. 

### 4.11. Electron Microscopy Tomography

Routine EM tomography and two-step correlative EM tomography were performed as described previously [23]. Briefly, 200-nm serial sections were cut (EM-UC6 ultra-microtome; Leica Microsystems) using a diamond knife (Diatome, Biel, Switzerland). Ribbons of sections were transferred to Formvar-coated, 1 × 2-mm-slot grids. Colloidal gold particles (10 nm) were placed on both surfaces of the grid to serve as fiducial markers for subsequent image alignment. Grids were imaged under transmission EM (Tecnai-20; ThermoFisher [previously FEI], Eindhoven. The Netherlands) operated at 200 keV. Single-axis tilt series were acquired automatically with images taken at 1° increments over ±65° about the orthogonal axes. Tomograms were calculated, analysed, and segmented using the IMOD software package. Initially, EM tomography was carried out at low magnification, with immediate examination of serial slices, leaving the sample in the microscope column. When the important detail was found, this area was examined at high (50,000×) magnification, with the high-resolution tomograms then made.

### 4.12. Focused Ion Beam Scanning EM

Focused ion beam scanning EM (FIBSEM) was performed according to the above-described protocol based on the use of reduced OsO_4_ and thiocarbohyrazide and embedding into Epon. The resin blocks were polished with a diamond knife and then trimmed to form a square pyramid of 2 mm × 2 mm. The blocks were evaporated with a 2 nm layer of gold and then placed into and examined in the scanning EM (Auriga; Zeiss, Oberkochen, Germany). Similar analysis was performed also using field emission (EMFEI Helios NanoLab 660 FEGSEM or G3) equipped with a scanning EM multi-detector and in-column detector, at accelerating voltage 2.0 kV [117]. Access to these was kindly provided by FEI Co. For all high-resolution EFSEM images, a primary beam energy of 2.0 kV was used, with a working distance of 1 mm, 3 ms dwell time, and tube bias of 140 V. A FIBSEM microscope (Auriga 60; Zeiss) with the Atlas3D software (FIBICS) also used to collect the three-dimensional data of two cells. Acquisitions were performed according to the manufacturer’s instructions.

Initially, the original polished surface was examined using back-scattered electrons at an accelerating voltage of 1.5 kV, and the zone of the hepatocyte containing the Golgi was selected. This area was coated with a 20-nm layer of platinum using a gas injector system, directly in the main specimen chamber of the SEM, to provide a smooth, conducting surface, and then a pyramid was trimmed. Secondary electron scanning images of the pyramid were typically recorded at accelerating voltages of 3 kV to5 kV in immersion lens mode. Next, the plane surface perpendicular to the platinum-coated surface was generated using a focused ion beam (gallium source) at 30 kV, and beam currents ranging from 0.5 nA to 7.0 nA, and then the 6-nm layers of sample within the zone of interest were milled within the new perpendicular surface. After each milling step, the newly formed surfaces were examined with back-scattered electron SEM, and the images were acquired. 

### 4.13. Sampling

In each set of experiments, we used six pairs of MatTek dishes (control [i.e., microinjection of an irrelevant protein or antibody, or transfection with irrelevant oligos] and experimental), with two randomly selected cells within the very centre of the Petri dishes examined. In all of the experiments, after seeding the cells, we used six pairs of samples composed of one randomly selected control and one randomly selected experimental Petri dish (or the very low edges of the rat). Every pair was further processed as a single sample under identical conditions. After embedding into Epon, vertical serial 1-µm sections through the cells of interest were made, and then on these vertical sections, the cross diameters of the nuclei were examined. The section where the diameter was maximal was re-embedded, and serial ultrathin vertical sections were obtained. The section where the width of the nucleus was maximal was considered as the central section, and the vertical axis was generated through the nuclear centre. Ultra-sectioning was organised in such a way that the control and experimental samples were embedded in one section block [118]. The control and experimental Petri dishes were glued and cut as one sample. In each dish, two cells were randomly selected: one in the very centre of the dish and the other in the peripheral part of the dish. Random representative images of these cells were examined. Two measurements (slices) were made in each cell. For quantification, vertical sections of cells were selected where four consecutive serial sections of a centriole were visible. Next, the normalised (or absolute) value for the experimental cup was calculated as a proportion (%) of its control. These proportions were used as units (N = 6) for calculation of means and standard deviations. 

### 4.14. Stereology

For stereological analysis, 50-nm Epon vertical sections or EM tomography were used. On serial sections, according to the heterokaryon assay, the pellets obtained after centrifugation of the cells were sectioned vertically (or perpendicular to the maximal cell width). A cell was considered as sectioned vertically when the ratio between the long and short axes of its nucleus was >3. The selection was then checked on serial sections. To estimate the proportion (%) of cells with the defined organelle, randomly selected sections of the cell were examined, passing through the centrosome area or the nucleus centre, and if the structure was found, this section of cells was considered as positive. Conversely, if, after careful examination, the organelle was not found, this section of the cell was considered negative. We examined three or four such sections taken from each control and experimental petri dish of all of the six pairs of experiments.

The Golgi complex was measured using the discretised version of the vertical rotator [119]. The number of organelles was estimated according to the rotator [120]. To count the absolute numbers of cisternal pores within Golgi stacks, pre-treated cells were used to depolymerise microtubules (33 µM nocodazole for 3 h). In serial vertical sections of the Golgi complex, the section where the Golgi complex had maximal width was selected, placed in the middle of the vertical axis, and then applied to the discretised rotator. 

To measure the surface volume density, a square test-grid was placed over three randomly selected cell areas of the cell of interest, with the numbers of test-points (P) inside an organelle were calculated, and the numbers of intersections (I) between the ER (or ERES compartments) and the test-grid were used to measure the surface density of the organelle membrane (Sv = 2∑I/d∑P, in µm^2^/µm^3^; where d is the real distance between the test-lines in the defined image). The absolute surface area was calculated by multiplying the surface volume density of the organelle of interest by the organelle volume [106]. The volumetric surface density (amount of the surface area in the unit of the volume) was measured according to Mironov et al. [106]. Briefly, vertical serial 1-µm sections were made through the cell of interest, and then the section where the nucleus diameter was maximal was re-embedded. Next, serial ultrathin vertical sections were obtained. The section where the width of the nucleus was maximal was considered as the central section, and the vertical axis was generated through the nuclear centre [120]. Then the absolute organelle volumes were measured using the discretised version of vertical rotator [119]. To measure the surface density in the organelle volume, the square test-grid was placed over three randomly selected areas of the cells of interest, and the numbers of test-points (P) inside an organelle were calculated, along with the number of intersections (I) between membranes of organelles and the test-lines (Sv org = 2∑I/d∑P in µm^2^/µm^3^; where d is the real distance between the test-lines corresponding to the situation within the section). The absolute surface areas were calculated by multiplying the surface volume density of the organelle of interest by the organelle volume [106,119]. Calculations of proportions (%), profiles and point counting were performed directly in the EM (Tecnai-20; FEI [now ThermoFisher Scientific]). The AnalySis software (ThermoFisher Scientific) was used for the measurements. 

### 4.15. Statistical Analysis

To test whether differences were significant (i.e., *p* < 0.05), Student’s *t*-tests, paired *t*-tests and non-parametric Mann–Whitney U tests were used. Normality of the datasets was assessed using Shapiro–Wilk normality tests. In the majority of cases, we used nonparametric Mann–Whitney U tests. A difference was considered significant when *p* < 0.05. The standard software package GraphPad Prism (Prism: Version 9.4.2) was used. Data are given as means ±standard deviation (SD). In the text, the words “differ”, “smaller”, or “higher” indicate that two values are significantly (*p* < 0.05) different.

## Figures and Tables

**Figure 1 ijms-23-03590-f001:**
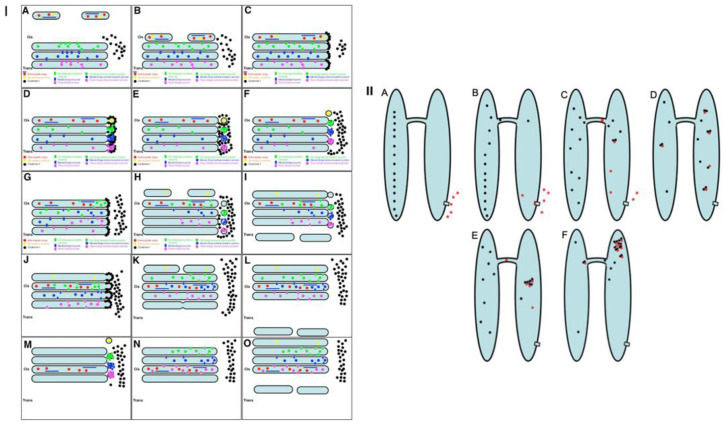
Schemes of intra-Golgi transport (IGT). ((**I**): (**A**–**O**)) Mechanisms of IGT according to the cisterna maturation–progression model of intracellular Golgi transport (CMPM; see Appendix A). (**A**) Two ER-Golgi carriers (EGCs) are shown in the upper part of the Figure. They contain mega-cargo (blue lines) and soluble secreftory cargo (red dots). (**B**) These carriers arrive on the opposite side of the Golgi complex. (**C**) The EGCs fuse and form a new cis-cisterna that contains these mega- and soluble cargoes. Coatomer I (black dots) forms coats on cisternal rims. (**D**) Formation of COPI-coated buds on rims of all of the Golgi cisternae. These buds are enriched in Golgi glycosylation enzymes corresponding to the positions of the cisternae on which they are formed. (**E**) Division of the copy-covered kidneys. During division, these buds turn into ‘bubbles’ enriched with Golgi enzymes and then undergo coat removal. (**F**) Fusion of COPI-dependent vesicles with proximal cisterna. Recycling of Golgi enzymes into the corresponding proximal cisterna. (**G**) Attachment of COPI to the rims of the cisternae. (**H**) Arrival of new ER-Golgi carriers and formation of a new cis-cisterna, as well as fission of the neck of COPI-coated buds and division of bubbles, and formation of an empty trans-cisterna. (**I**) Merging of carriers to form a new cis- cisterna. Fusion of COPI-dependent vesicles with proximal cisternae and formation of two carriers after the Golgi from an empty trans-cisterna. (**J**–**O**) Two consecutive rounds of the same process that occurs distally along the Golgi stack. As a result, the cis- cisterna containing the mega-cargo (blue lines) and soluble cargo (red dots) appears on the trans-position within the stack. ((**II**): (**A**–**F**)) Mechanisms of concentration of soluble cargoes according to the symmetric KARM (the carrier maturation model). Black dots, soluble cargo; red dots, proton pumping inside the distal compartment. The soluble cargo can diffuse in both directions through a thin tube connecting the two compartments. In the bottom part of the distal compartment, there is a proton pump ((**A**); square), which moves protons into the lumen of the distal compartment (**B**). When a proton attaches to the cargo, the cargo molecule tends to form aggregates (**C**,**D**). Aggregates become larger than the original molecule and cannot move through the tubes in the retrograde direction. After several cycles of such transformation (**E**), the concentration of the cargo in the distal compartment becomes higher than in the proximal compartment (**F**).

**Figure 2 ijms-23-03590-f002:**
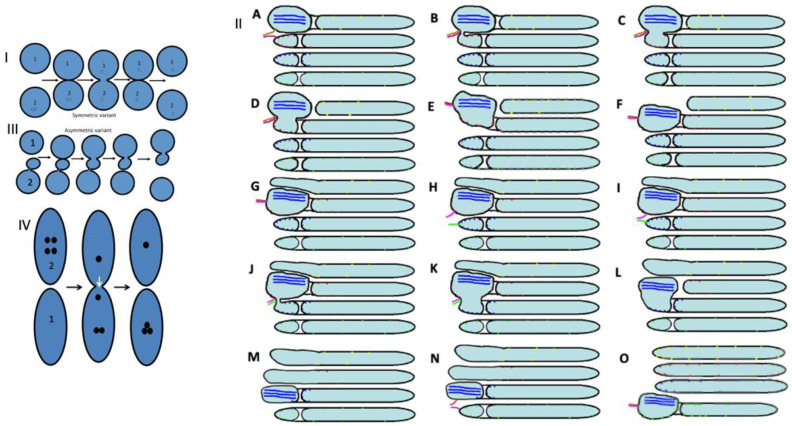
Schemes of the kiss-and-run models of intracellular transport (**I**–**IV**) and (**A**–**O**). Adapted from Mironov and Beznoussenko [38]. (**I**) The symmetrical KARM poses that after fusion of the membranes of these two compartments (central image), cargo (black dots inside the ring 2) would diffuse into the lumen of compartment 1. The concentration of cargo would be similar in both compartments. (**II**) The asymmetric KARM poses that distal compartment 2 is composed of the main part and smaller parts where the cargoes are concentrated. These two parts are connected by thin tubules. Thus, the KARM suggests that the compartments initially fuse, and then for some reason, the tubules undergo fission. For instance, these tubules can be broken easily when lipids from the distal compartment diffuse into them. An additional demand for the asymmetrical KARM is the necessity for a mechanism responsible for the change in the cargoes in such a way that this would induce a greater formation of temporal aggregates by the cargoes. These temporally existing aggregates would be not able to diffuse through the thin tubules backwards. (**A**–**O**) Function of the asymmetrical KARM within the Golgi stack (see Appendix A). (**A**) Formation of the SNARE complex composed of V- and T-SNAREs (brown and magenta lines, left) between the mega-cargo (blue lines) containing cisternal distensions and the rim of the distal cisterna. (**B**) Fusion between the distensions and the corresponding rim. (**C**–**F**) Integration of the distension into the distal cisterna. (**G**) Elongation of the first cisternae. (**H**,**I**) Replacement of the SNAREs and formation of the new SNARE complex composed of another set of SNAREs (red and green line). (**J**) Fusion of the cargo-containing distension situated within the second cisterna with the rim of the third cisterna containing a pore. (**K**–**O**) Additional rounds of fusion/fission processes.

**Figure 3 ijms-23-03590-f003:**
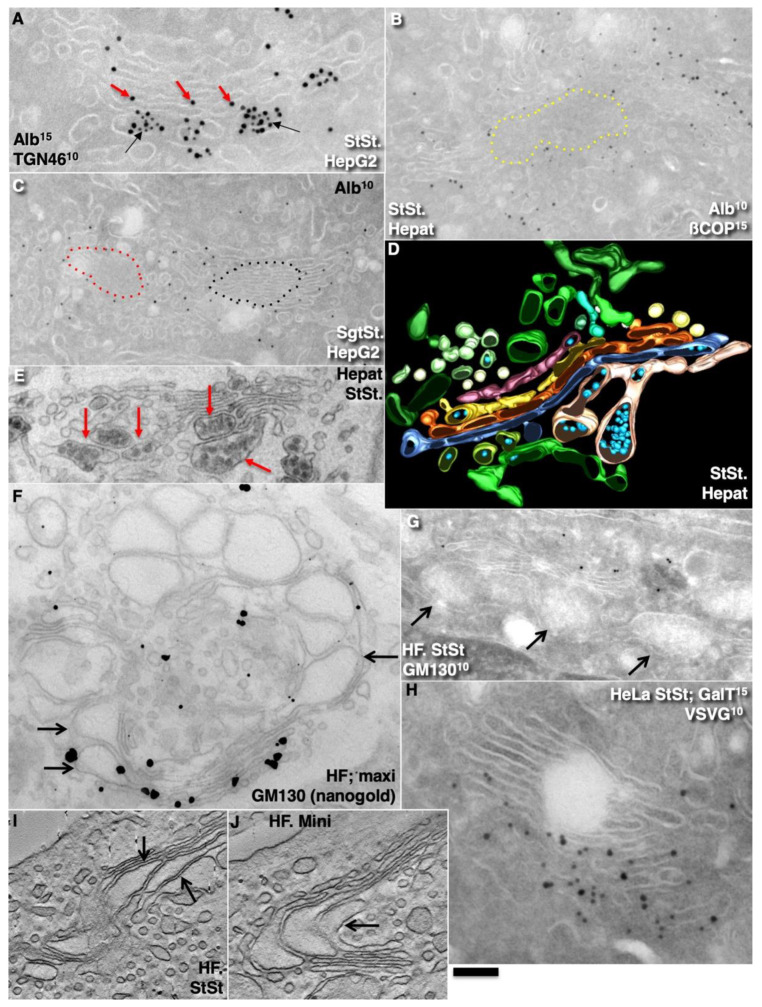
Representative images show enrichment of cargoes at the trans-side of the Golgi complex. The type of cell, protein labelling and the state of transport (steady-state [StSt]; mini-wave [Mini]) are indicated on the images. (**A**) Enrichment of albumin (red arrows) in the last medial cisterna and in TMC (black arrows). Tokuyasu cryo-section. (**B**,**C**) Enrichment of albumin in the cisternal lumen near the rims. Tokuyasu cryo-section. (**D**,**E**) Enrichment of VLDL in one cisternal distension of a hepatocyte. (**D**) Three-dimensional model of EM tomography. (**E**) Routine transmission EM. (**F**) High concentration of PCI distensions (arrows) in the last medial cisternae of human fibroblasts. Enhanced nanogold. (**G**) Enrichment of procollagen I (PCI) distensions (arrows) at the trans-side of the Golgi. Tokuyasu cryo-section. (**H**) Enrichment of VSVG labelling (10 nm gold) at the trans-side of the Golgi stack. Tokuyasu cryo-section. (**I**) Enrichment of PCI distensions (arrows) within the last medial cisternae in human fibroblasts at steady-state. EM tomography. (**J**) Enrichment of PCI distensions (arrows) in the medial cisternae at the trans-side of the Golgi complex of human fibroblasts under a mini-wave. EM tomography. Scale bars: 130 nm (**A**,**D**,**H**); 200 nm (**B**,**C**,**E**); 240 nm (**F**,**G**,**I**,**J**).

**Figure 4 ijms-23-03590-f004:**
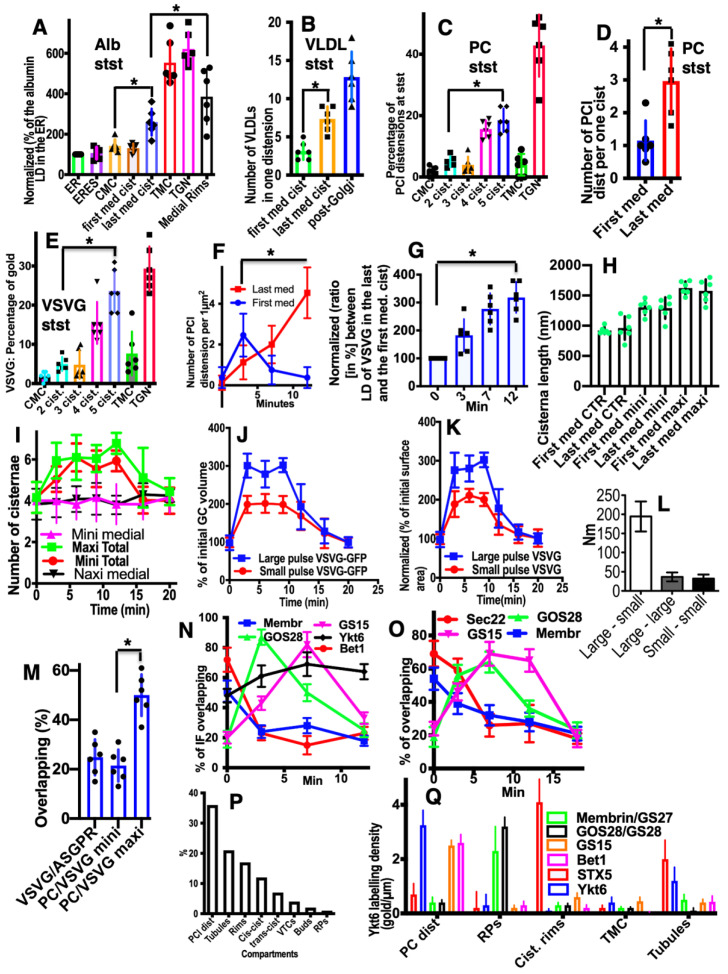
Quantification of data. (**A**–**E**) Steady-state (stst). Normalised (versus the ER) gold labelling of albumin (**A**) and VSVG (**E**). Normalised (versus the first medial cisterna) number of VLDLs in one cisternal distension (**B**). Normalised number of procollagen I-containing distensions in one cisterna (**C**,**D**). (**A**) Gold labelling for albumin in the last medial cisterna and the *trans*-most cisterna is significantly higher than that in the first medial cisterna (*p* < 0.05). (**B**) The number of VLDLs in one cisternal distension of the last medial cisterna is significantly higher than that in the first medial cisterna (*p* < 0.05). (**C**) Distribution of PCI distension within the Golgi complex. Its proportion in the last medial cisterna is significantly higher than that in the first medial cisterna (*p* < 0.05). (**D**) The numerical density of PC distensions in the last medial cisterna is significantly higher than that in the first medial cisterna (*p* < 0.05). (**E**) Distribution of VSVG gold labelling within the Golgi complex. The numeric density of gold in the last medial cisterna is significantly higher than in the first medial cisterna (*p* < 0.05). (**F**) The mini-wave. The number of PCI distensions per 1 µm^2^ of cisterna. Their maximal number in the last medial cisterna is significantly higher than in the first medial cisterna (*p* < 0.05). (**G**) Labelling for VSVG becomes significantly enriched in the last medial cisterna (*p* < 0.05). (**H**) The length of the first and last medial cisternae do not differ both at steady-state (CTR) and during synchronous IGT (mini and maxi-waves). (**I**) During IGT (according to the mini-wave and maxi-wave protocols), the number of medial Golgi cisternae does not change, whereas two additional cisternae (*cis*-most and *trans*-most cisterna) are attached. (**J**,**K**) Dynamics of Golgi volume (**J**) and surface area of Golgi compartments (**K**) during the synchronous IGT of VSVG-GFP according to the mini-wave (red line) and maxi-wave (blue line) protocols. During IGT, the volume of the Golgi complex and the surface area of the Golgi compartments (**K**) significantly increased (*p* < 0.05), but this depended on the amount of cargo transported. (**L**) Distance between large (VSVG) and small (ManII) gold particles is significantly higher than between large gold particles or between small gold particles (*p* < 0.05). (**M**) Overlapping of immunofluorescence (IF) labelling between PC and VSVG when maxi-wave synchronisation was applied is significantly higher than when the mini-wave was used (*p* < 0.05). During IGT synchronised according to the cycloheximide (CHM)-15-CHM protocol, overlap between ASGPR and VSVG was low, similar to that after the mini-wave PCI and VSVG IGT. (**N**,**O**) Dynamics of co-localisation between PCI (**N**) and VSVG (**O**) with different Golgi SNAREs during the synchronised IGT, according to the CHM-15-CHM protocol. (**N**) Overlapping of IF between PCI and different Golgi SNAREs during IGT. PCI lost (*p* < 0.05) its co-localisation with Bet1 and then acquired co-localisation with GOS28 and GS15. (**O**) Overlapping of IF between VSVG and different Golgi SNAREs during IGT. VSVG lost (*p* < 0.05) its co-localisation with Sec22 and membrane and then acquired co-localisation with GOS28 and GS15. (**P**) distribution of immune-EM labelling for Ykt6 during IGT of PCI. Mini-wave. Ykt6 is depleted in round profiles (RPs) and enriched over PCI distensions (PCI dist.). (**Q**) Distribution of labelling of different Golgi SNAREs over the Golgi compartments at 4 min after release of the transport block of PCI. Mini-wave. Membranes of PCI distensions contain Ykt6, GS15 and Bet1. RPs are enriched in membrane and GOS28. Cisternal rims are enriched in syntaxin-5 (STX5). * *p* < 0.05.

**Figure 5 ijms-23-03590-f005:**
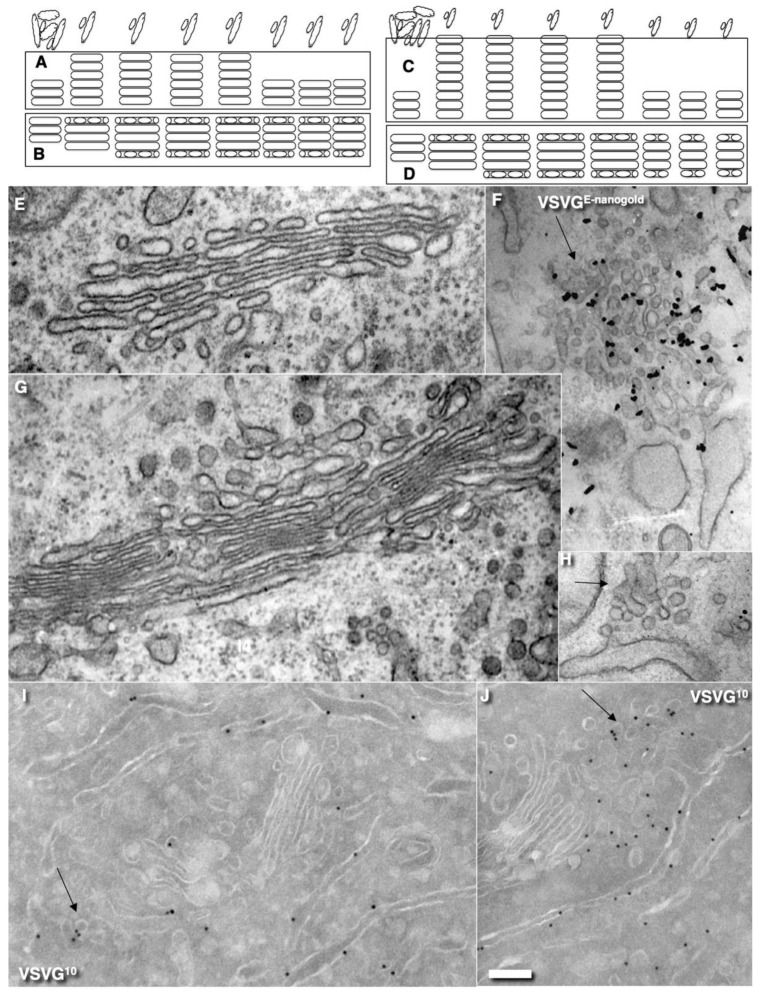
Dynamics of Golgi compartments during synchronous IGT. (**A**–**D**) Schemes of predictions related to the numbers of medial Golgi cisternae during IGT derived from the CMPM (**A**,**C**) and the KARM (**B**,**D**) during the mini-wave (**A**,**B**) and maxi-wave (**C**,**D**). According to the KARM, the number of medial Golgi cisternae remains constant both before and after the release of the transport block and does not depend on the amount of VSVG moving through the Golgi complex. The volume of the Golgi complex (**A**) and the surface area of the Golgi compartments (**B**) depends on the amounts of cargo transported (indicated in upper row). The KARM predicts that during the maxi-wave, the cisterna length would be higher than during the mini-wave. (**E**–**J**) Representative EM images. (**E**,**G**,**H**) Routine EM. (**F**) Enhanced gold pre-embedding IEM (E-nanogold). (**I**,**J**) Tokuyasu cryo-sections. (**E**) The Golgi complex was small just before the release of the transport block (**F**,**G**), whereas the volume of the ERES was high (**F**). The Golgi volume increased after arrival of the cargo (**G**), whereas the volume of the ERES became small (**H**). Then the Golgi volume decreased again (**I**), whereas the volume of ERES became higher (**J**) if the delivery of the cargo was organised as a wave. Black arrows show ERES. Scale bars: 200 nm (**C**,**E**,**F**,**G**); 290 nm (**D**). Quantified in Figure 4H–K.

**Figure 6 ijms-23-03590-f006:**
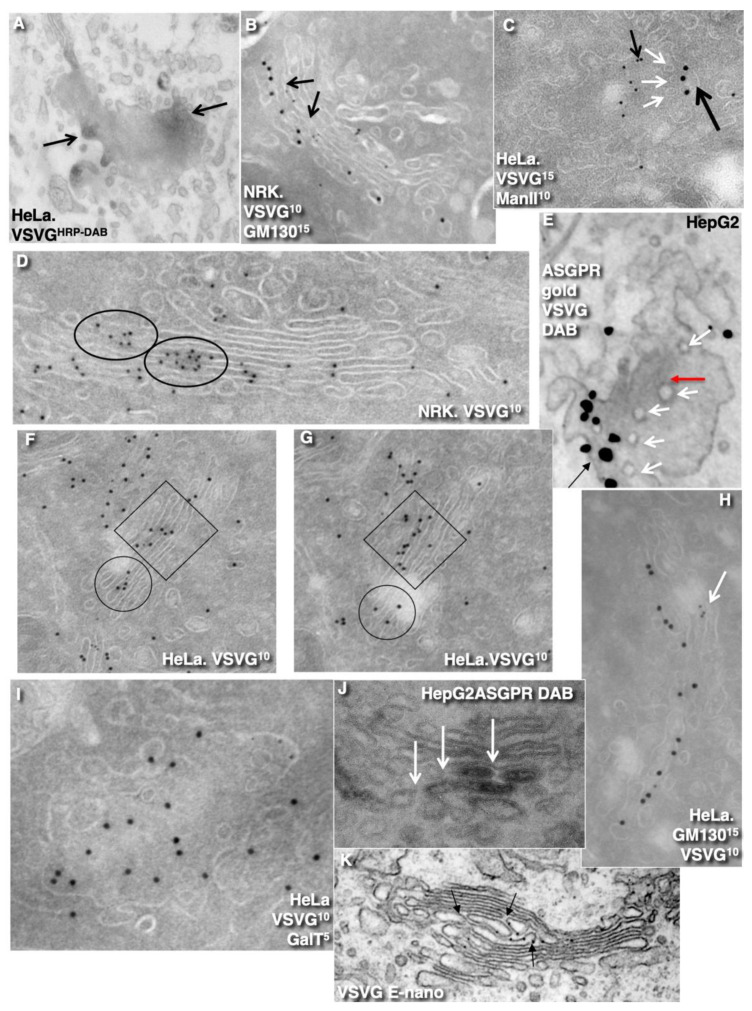
During intra-Golgi transport of small amounts of VSVG and ASGPR (CHM-15-CHM protocol), their distinct domains are formed. (**A**,**E**,**J**) HRP-DAB reaction. (**B**–**D**,**F**–**I**) Tokuyasu cryo-sections. (**F**,**G**) Serial Tokuyasu cryo-sections. (**E**,**K**) Enhanced-nanogold. (**A**–**I**,**K**) Domains of VSVG and ASGPR (**E**,**J**) at 5 min after release of the transport block. Clusters (arrows) of 10-nm gold particles as indication of VSVG domains along the Golgi stacks. (**C**) Domains of VSVG (large black arrow) and ManII (small black arrows). White arrows indicate pores around VSVG domain. (**E**) Domains of ASGPR (gold; black arrow) and VSVG DAB (red arrow) are distinct. White arrows indicate pores around cargo domains. (**J**) Domain of ASGPR (DAB labelling) is surrounded with pores (white arrows). Markers and gold size are indicated on images. Circles and squares indicate VSVG domains. Scale bars: 280 nm (**F**–**L**); 200 nm (**M**,**Q**).

**Figure 7 ijms-23-03590-f007:**
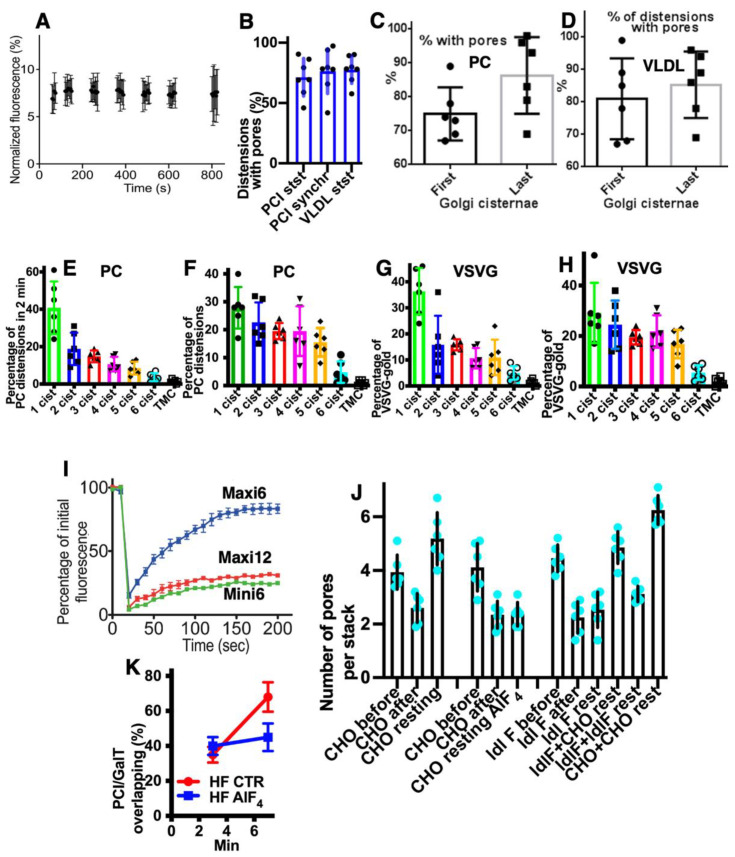
Quantification of the results. (**A**). Fluorescence intensity of VSVG dots does not change during intra-Golgi transport. (**B**) Proportion of PCI and VLDL distensions separated with the pore row is high. (**C**,**D**) Proportions of cisternal distensions surrounded with pores. Pores surround cisternal distensions with PCI (**C**) and VLDL (**D**) in both the first and last Golgi cisternae. (**E**–**H**) Distribution of gold labelling for PC distensions (**E**,**F**) and VSVG (**G**,**H**) at 2 min after release of transport block, according to the mini-wave (**E**,**G**) and maxi-wave (**F**,**H**) protocols. Labelling densities of gold and distensions in the first medial cisterna in (**E**,**G**) are significantly higher than in the last medial cisterna (*p* < 0.05), whereas this difference is less in (**F**,**H**). (**I**) Fluorescence recovery (dynamics of the ratio between fluorescence intensity between the bleached zone and ER). Synchronisation protocols and times after the release of the temperature blocks are indicated in the images. After the maxi-wave synchronisation, the recovery is greater at 6 min than those during the mini-wave and at 12 min after release using the maxi-wave. (**I**,**J**) Inhibition of the COPI function decreases the rate of pore restoration decrease. In CHO and ldlF cells, the COPI function was inhibited by heating cells to 40 °C for 2 min. In CHO cells, the bar after is lower than the bar resting. In ldlF cells, this difference is not significant. In heterokaryons composed of CHO cells, the restoration is high. In the heterokaryons composed of ldlF cells, it remains low. In heterokaryons composed of CHO and ldlF cells, the speed of the pore restoration is higher than in ldlF cells. (**K**) During intra-Golgi transport, co-localisation between PCI and GalT increases in control (CTR) cells but is inhibited in cells treated with AlF_4_.

**Figure 8 ijms-23-03590-f008:**
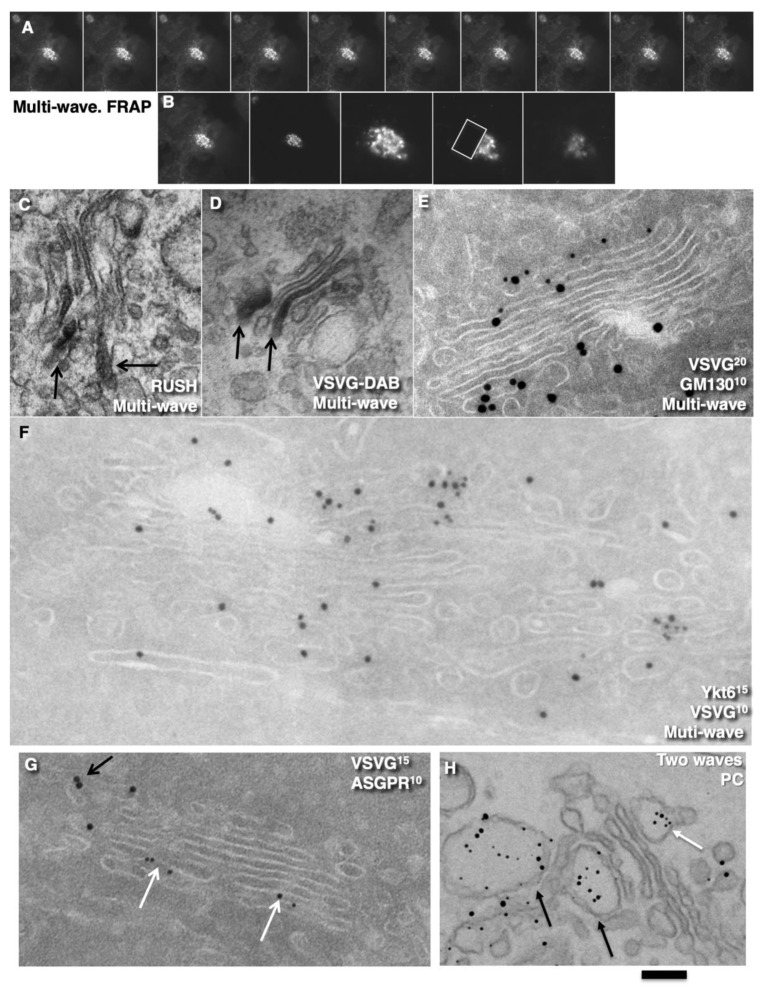
Multi-wave synchronization protocol. HeLa (A-F), HepG2 (G) cells, and human fibroblasts (H) were subjected to multi-wave synchronisation protocols (see Methods). (**A**,**B**) Fluorescent microscopy. (**A**) Kymographs of VSVG-GFP dynamics within the Golgi mass. Carriers moved in both directions. (**B**) Kymographs of the same Golgi mass (two first images), higher magnification (middle image), and subsequent bleaching of half of the Golgi complex. (**C**) Immuno-peroxidase labelling for RUSH-TNF-α in different Golgi cisternae (arrows) after synchronisation according to multi-wave intra-Golgi transport (see Methods). (**D**) After the multi-wave protocol, VSVG is present in different cisternae (immuno-peroxidase labelling; DAB). (**E**) Tokuyasu cryo-sections. VSVG is present in the first near Golgi cisterna (GM130) and the last medial Golgi cisternae. (**F**) Tokuyasu cryo-sections. Distinct domains of VSVG (10-nm gold) contain enriched Ykt6 (15-nm gold). (**G**) Tokuyasu cryo-sections. In HepG2 cells infected with tsVSV and synchronised according to the CHM-15-CHM protocol, VSVG and ASGPR form distinct domains at 5 min after release of the transport block. (**H**) In human fibroblasts, after arrival of the second wave of PC, the first portion (white arrow) was already at the trans-side (black arrows) of the stack. Enhanced nano-gold. Scale bars: 8 µm (**A**); 6 µm (**B**); 210 nm (**E**); 250 nm (**C**,**H**); 85 nm (**F**); 170 nm (**G**).

**Figure 9 ijms-23-03590-f009:**
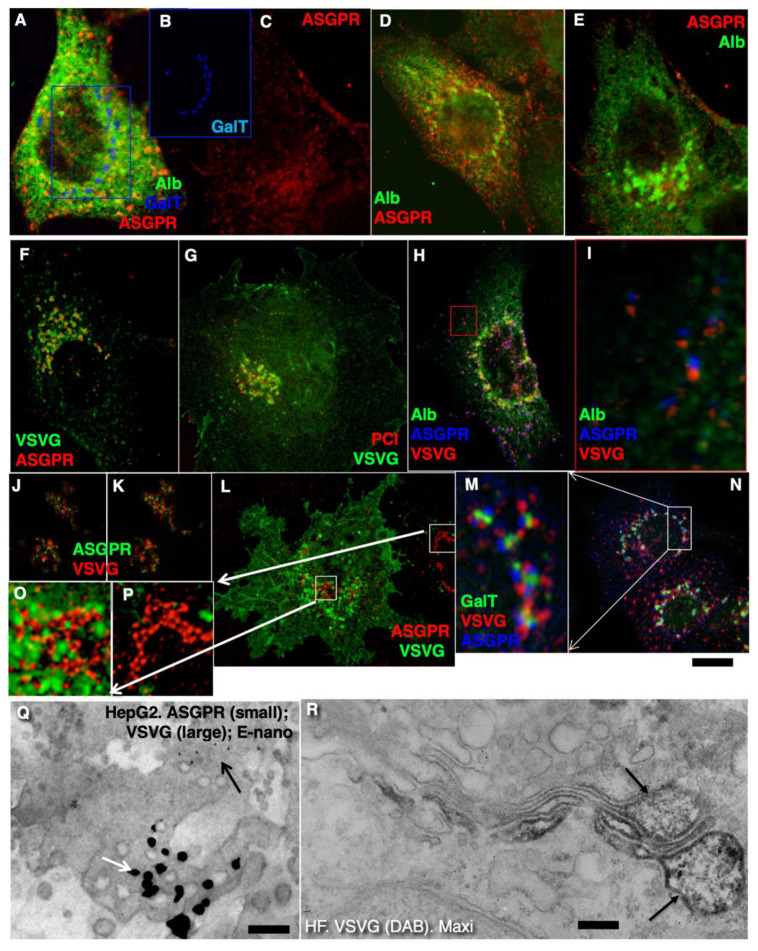
During intra-Golgi transport, different membrane cargoes form distinct domains. (**A**–**C**) Fluorescence microscopy. HepG2 cells are shown after the 15 °C temperature block. Albumin (green) is in the ER, ASGPR (red; (**C**)) is in the ER exit sites. (**B**) Labelling for GalT (blue) in the Golgi complex appears to be centrally fragmented. (**D**) At 2 min after release of the 15 °C temperature block, part of the albumin is in the Golgi complex, whereas ASGPR remains in ER exit sites. (**E**) Labelling for ASGPR (red) and albumin (green) at 4 min after release of the 15 °C temperature block. Albumin (green) is concentrated within the Golgi complex. (**F**,**H**–**P**). Fluorescence microscopy. In HepG2 cells infected with tsVSV and synchronised according to the CHM-15-CHM protocol, VSVG and ASGPR form distinct dots at 5 min after release of the block. (**G**) In human fibroblasts infected with tsVSV, stimulated to synthesise PCI and incubated at 32 °C, PC and VSVG form distinct dots. (**Q**) Double silver enhancement. Distinct aggregates of different gold particles. (**R**) When the maxi-wave protocol was applied, VSVG (DAB) was present in the membrane surrounding PCI distensions (arrows). Scale bars: 2 µm (**A**–**G**); 4 µm (**H**); 5 µm (**L**,**N**); 180 nm (**C**,**Q**); 280 nm (**R**).

**Figure 10 ijms-23-03590-f010:**
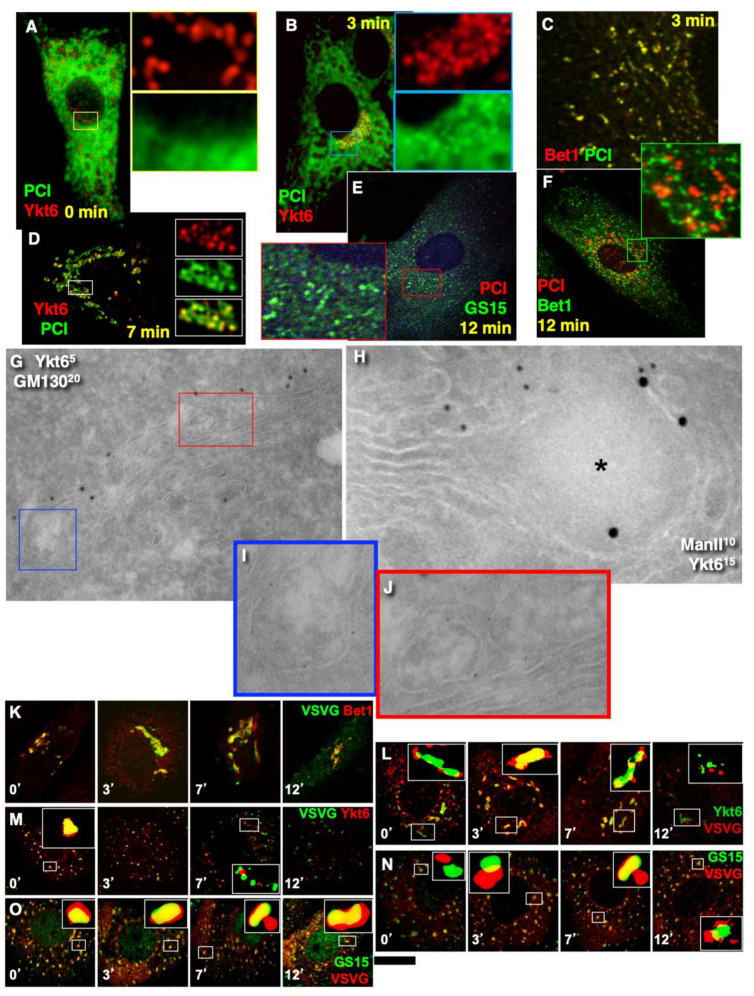
Enrichment of SNAREs in membranes surrounding the cargo domains. (**A**–**J**) Human fibroblasts. (**K**–**O**) HeLa cells. (**A**–**F**,**K**–**O**) Fluorescence microscopy. (**G**–**J**) Tokuyasu cryo-sections. (**A**–**O**) Mini-wave. (**A**) Before the release of the transport block, Ykt6 was a little enriched at ER exit sites. (**B**) At 3 min after the release of the transport block, PCI formed dots where Yklt6 became enriched. (**C**) PCI dots co-localised with Bet1. (**D**) At 7 min after re-initiation of intra-Golgi transport, PCI and Ykt6 dots were co-localised. (**E**,**F**) At 12 min after re-initiation of intra-Golgi transport, PCI dots lost their co-localisation with Ykt6 and Bet1 but acquired co-localisation with GS15. (**G**–**J**) Enrichment of Ykt6 in membranes surrounding PCI distensions (blue, red boxes). (**I**,**J**) Enlargement of areas inside the blue and red boxes in (**G**). (**K**,**L**) VSVG-containing dots acquired Bet1, then Ykt6, and finally GS15, whereas Bet1 disappeared in these dots. (**M**–**P**) Fluorescence microscopy. Ministacks (pre-treatment of cells with 33 µM nocodazole for 3 h). Times of IGT are indicated on images ((**K**–**O**); quantified in Figure 2N,O). Scale bars: 4 µm (**A**–**F**,**K**–**P**); 130 nm (**G**); 90 nm (**H**).

**Figure 11 ijms-23-03590-f011:**
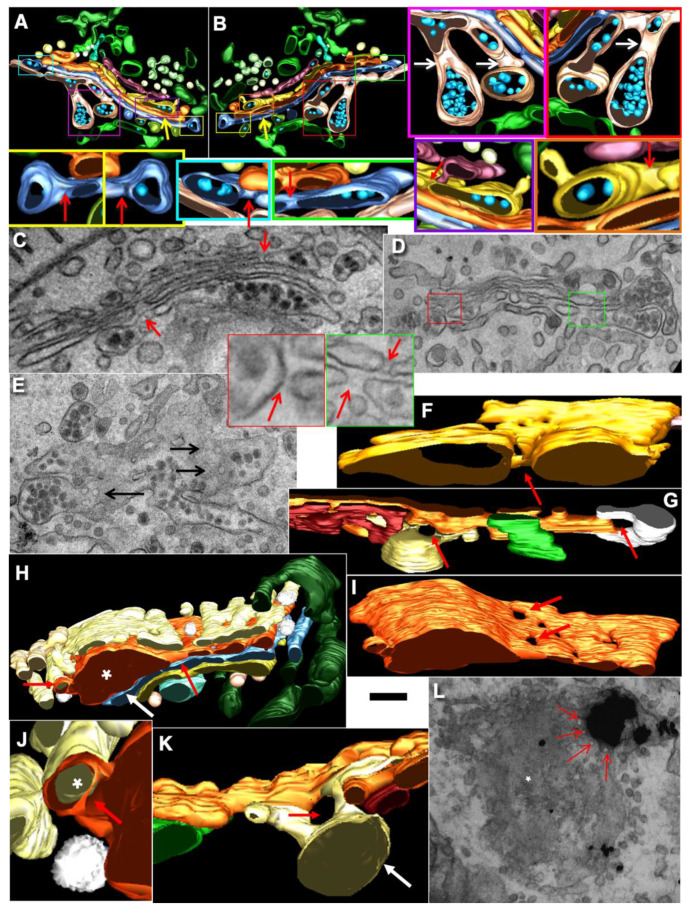
At steady-state (**A**–**E**) and during the mini-wave protocol (**F**–**L**), cisternal distensions filled with VLDLs (**A**–**E**) or procollagen-I (**F**–**L**) were separated from the rest of the Golgi cisternae by rows of pores. (**A**–**B**,**F**–**K**) EM tomography. (**C**–**E**,**L**) Representative transmission EM images. (**A**,**B**) Three-dimensional models of the Golgi complex of hepatocytes shown from opposite views (shown in Figure 3D). Green, ER. Several boxes with different colours of their borders were enlarged and demonstrated pores separating distensions filled with VLDLs. Red arrows, pores; yellow arrows, medial cisternae; blue spheres, VLDLs. (**C**,**D**) Pores (red arrows) between VLDL distensions and the rest of Golgi cisternae. Below: Enlargements of the areas inside the red and green-bordered boxes in (**D**). (**E**) Tangential section of Golgi cisternae with pores (black arrows). (**F**–**K**) Three-dimensional models that show pores (red arrows) between cisternae and PCI distensions. White asterisks in (**H**) and (**J**) indicate the lumen of cisternal distensions. (**J**) Red arrow, continuity between PC distension and another cisterna. (**L**) Tangential section of the Golgi stack. White asterisk indicates Gollgi cisterna. Scale bars: 210 nm (**A**,**B**); 120 (**C**,**F**–**I**,**K**); 170 nm (**D**,**E**); 240 nm (**L**).

**Figure 12 ijms-23-03590-f012:**
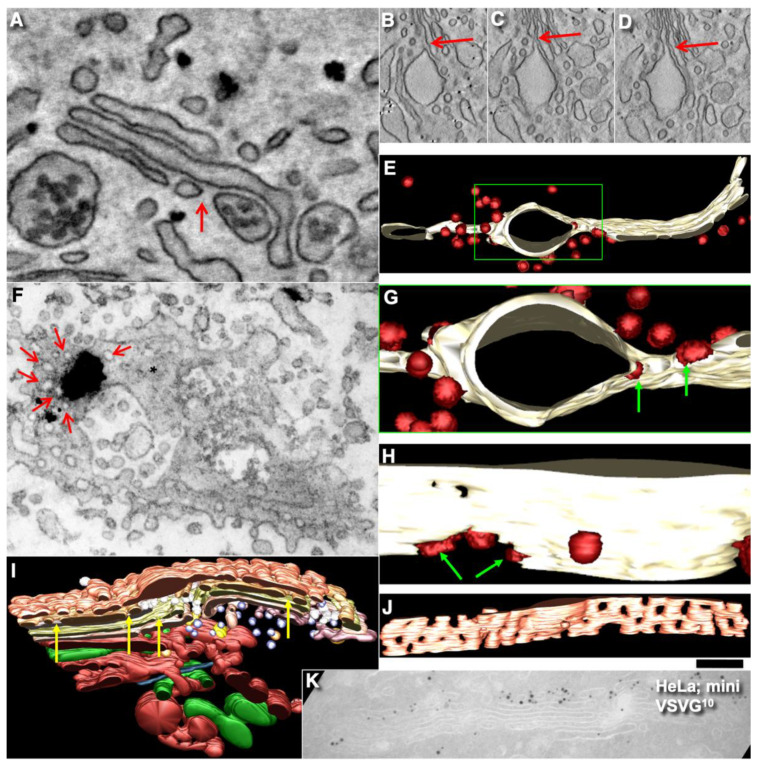
Pores separate cisternal distensions, which are filled with VLDLs (**A**) in hepatocytes and procollagen I in human fibroblasts (**B**–**J**). (**A**–**D**) All distensions are separated from the rest of the cisternae by pores (red arrows) around cisternal distensions. (**E**,**G**,**H**) Three-dimensional model of the Golgi cisterna during the mini-wave. Green arrows, COPI-coated buds (red) on the rim of the pore near the distension. (**F**) Mini-wave protocol. Tangential section of the medial Golgi cisterna. Immuno EM, enhanced nanogold particles show PC. Red arrows, pores surrounding PC aggregate (black blob) in the section. Asterisks indicate solid parts of Golgi cisternae. (**I**,**J**) Three-dimensional model of the Golgi complex with the PC-containing cisterna distension in the perforated cis-most cisterna at 2 min after the transport block. Yellow arrows, many pores in the medial cisternae. (**K**) HeLa cells. Mini-wave of VSVG. Low level of penetration of VSVG into the Golgi stack. Scale bars (nm): 165 (**A**); 420 (**B**–**D**); 240 (**E**,**F**); 75 nm (**H**); 280 (**I**,**K**).

**Figure 13 ijms-23-03590-f013:**
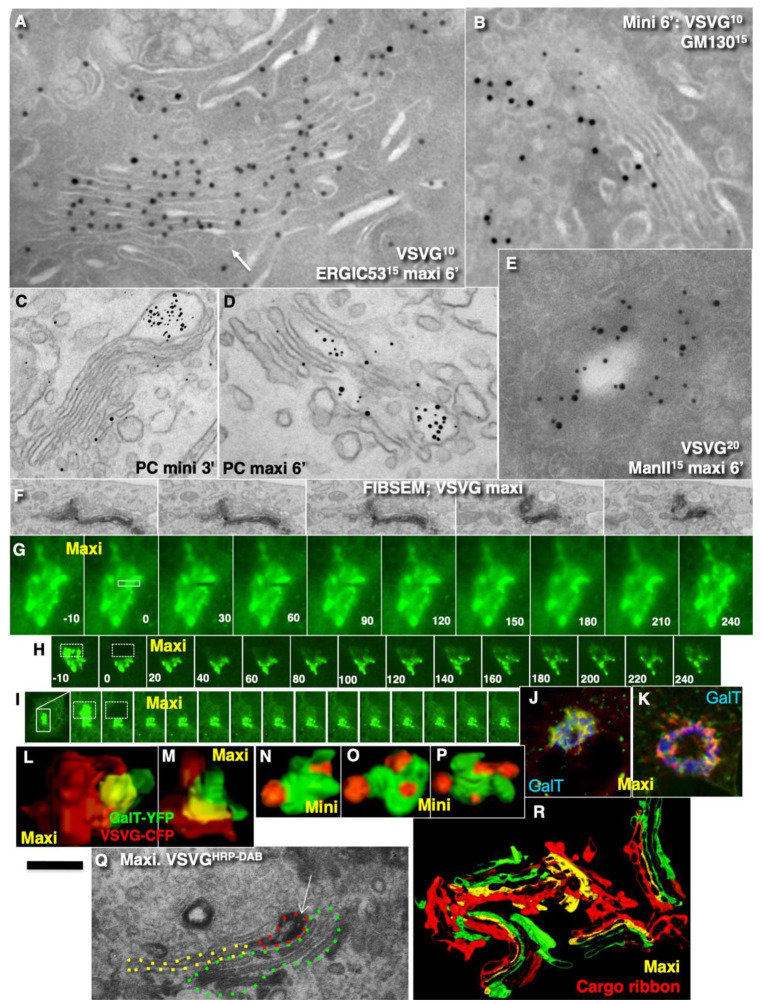
The pattern of intra-Golgi transport depends on the amount of cargo transported. (**A**,**B**,**E**–**R**) Hela cells. (**C**,**D**) Human fibroblasts. Protocols of cargo synchronisation are shown in images, as the mini-wave (**B**,**C**,**N**–**P**) and the maxi-wave (**A**,**D**–**M**,**Q**,**R**). (**A**,**B**,**E**) Tokuyasu cryo-sections. HeLa cells were transfected with VSVG-GFP (10-nm gold immunolabelling) and subjected to the maxi-wave and mini-wave protocols of cargo synchronisation. When a large amount of VSVG-GFP was transported (**A**), this cargo penetrated up to the trans-most cisterna (white arrow). Intermixing of VSVG-GFP and ManII was higher than after the mini-wave ((**E**); compare with Figure 6C). During the mini-wave (**B**), the penetration of VSVG was lower (quantified in Figure 7G–H). (**C**,**D**) Enhanced nanogold. Human fibroblasts were stimulated to synthesise PCI and then subjected to the mini-wave (**C**) and maxi-wave (**D**) protocols of PCI synchronisation. During the maxi-wave, the penetration of PCI was deeper (quantified in Figure 7E,F). (**F**) Focused ion beam scanning EM (FIBSEM) analysis. Serial images show that the immune–EM labelling for VSVG–GFP is not interrupted and forms a ribbon. (**G**–**I**) Fluorescence recovery after photobleaching (FRAP). HeLa cells were transfected with VSVG–GFP and then subjected to the maxi-wave protocol (see Methods). Then, the whole cell less half of its Golgi complex was bleached, and the FRAP was examined at 4 (**G**), 8 (**H**) and 12 (**I**) min after the initiation of intra-Golgi transport. Kymographs. Images were taken at times shown in (**A**,**B**) and every 30 s (**G**) or every 20 s (**H**,**I**). Fast FRAP was observed in (**G**,**B**). Lower FRAP was observed in (**H**). Very low FRAP was observed in (**I**). Data are quantified in Figure 7I. (**J**,**K**) Immunofluorescence. Overlapping of Golgi ribbon filled with VSVG-GFP before (green) and after bleaching and refilling (red). There is a high level of overlap. Significant areas of yellow colour suggest that the second portion of VSVG-GFP moved along the same structures where the first portion of VSVG-GFP was already present. The yellow zones show low co-localisation with GalT (blue). (**L**–**P**) Immunofluorescence. Three-dimensional reconstruction of Golgi fragments visible at the immunofluorescence level after synchronisation of VSVG according to the maxi-wave (**L**,**M**) and mini-wave (**N**–**P**) protocols. One large cargo sphere is formed in the first case, and several small spheres are formed in the second case. (**Q**) Immuno-EM labelling for VSVG with DAB reaction. The contour of the DAB-positive structures is traced with red dashed lines, the contours of the empty zones of the same cisternae are contoured with a yellow dashed line, and the other cisternae were contoured with green dashed lines. (**R**) Three-dimensional models of the Golgi cargo ribbons. Red structures form ribbons connecting different Golgi stacks. When a large amount of VSV-GFP was transported, these membranes formed a continuous ribbon. Previously we demonstrated that ManII-positive compartments of the Golgi form a ribbon [19]. Scale bars: 180 nm (**A**,**B**) 240 nm (**C**,**D**); 190 nm (**E**); 1.3 µm (**F**); 5 µm (**G**); 12 µm (**H**,**I**); 3 µm (**J**,**K**); 250 nm (**Q**); 450 (**R**).

**Figure 14 ijms-23-03590-f014:**
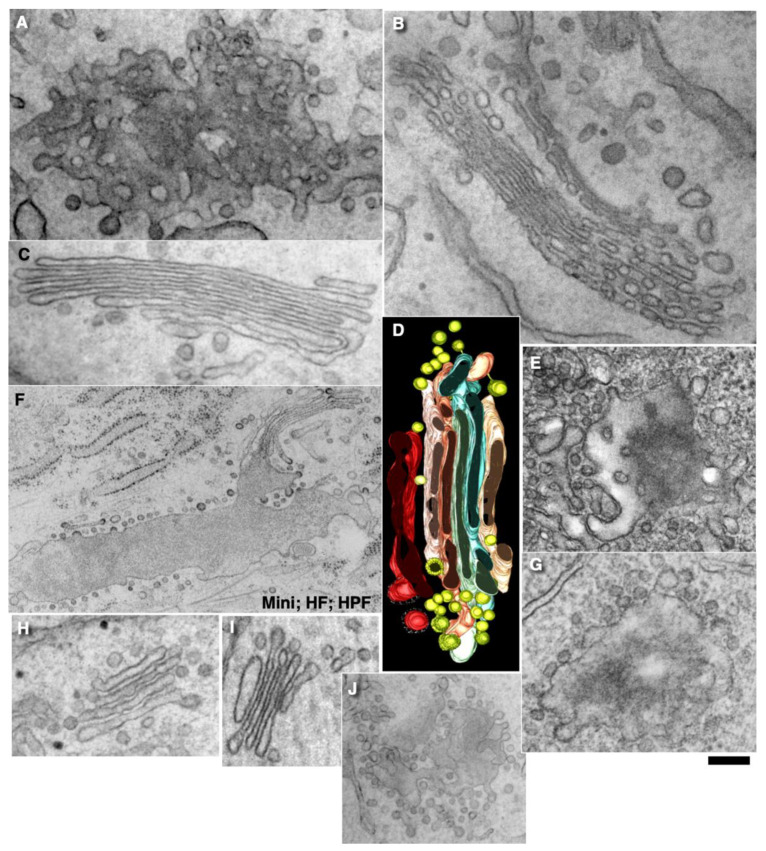
Dynamics of pores during intra-Golgi transport and role of COPI in the generation of cisternal pores. (**A**,**B**) Before the release of the transport block, according to the CHM-15-CHM synchronisation protocol, the numerical density of the pores over all of the Golgi cisternae was high. (**C**,**D**) Disappearance of cisternal pores after the passage of VSVG. (**F**) High-pressure freezing. Human fibroblasts. Disappearance of cisternal pores after passage of PCI through the Golgi. (**G**) Addition of AlF_4_ blocked restoration of the cisterna pores after the passage of VSVG. (**H**,**J**) Restoration of the numerical density of pores in the first medial cisterna after intra-Golgi transport and its inhibition for 25 min was blocked in heated ldlF cells. (see Methods). (**I**) Ministacks (3 h of nocodazole pre-treatment) after the passage of VSVG. There were no pores in the medial cisternae and no CMC and TMC. (**J**) Impairment of restoration of the cisternal pores in the ldlF cell heated to 40 °C for 2 min. Scale bars: 160 nm (**A**,**D**,**G**); 175 (**B**,**H**,**I**); 210 nm (**C**,**E**); 415 nm (**F**); 260 nm (**J**).

**Figure 15 ijms-23-03590-f015:**
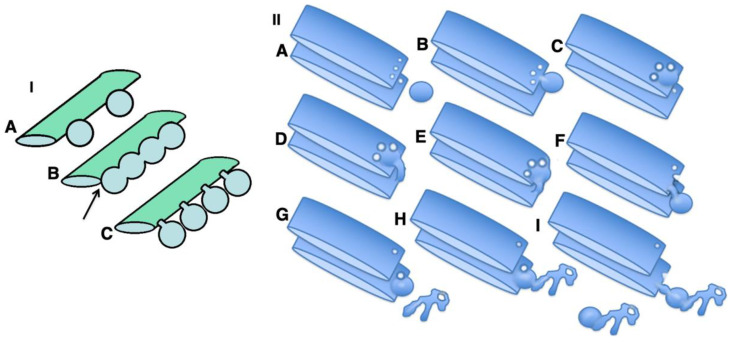
Scheme demonstrating the mechanism of IGT according to the KARM. (**I**). Formation of the cargo ribbon. (**A**) Arrival of two cargo domains. (**B**) Arrival of new domains and their fusion, leading to the parallel ribbon. (**C**) Budding of the cargo domains from the Golgi cisternae. (**II**). (**A**) The asymmetric KARM (carrier maturation) poses that a cargo domain (sphere in the upper-left image fuses with the cisternal rims containing several pores. This is shown in (**B,C**). (**D**,**E**) The cargo domain fuses with the distal cisterna with the tubule. (**F**) Break of the tubules within the zone where pores were situated. (**G**) Fission of the tubule connecting the cargo domain and the proximal cisterna. (**H**,**I**) Arrival of the trans-Golgi network, its fusion with the cargo domain, and the consecutive fission of the tubule connecting the cargo domain and the distal cisterna.

**Table 1 ijms-23-03590-t001:** Comparisons of predictions derived from the cisterna maturation–progression model (CMPM) and kiss-and-run model (KARM), and experimental results.

Observation	Model		
	CMPM	KARM	Experiment
Domain formation	Neutral	Desired	Shown
Increased concentration of SNAREs in the membrane of cargo domains	Neutral	Desired	Shown
Pores around cargo domains	Neutral	Desired	Shown
Cargo enrichment at the *trans*-side of the Golgi complex	Prohibitive	Possible	Shown
Stability of the number of medial cisternae during intra-Golgi transport	Prohibitive	Desired	Shown
Increased concentration of Golgi enzymes in COPI vesicles	Desired	Neutral	Not shown
Complete isolation of Golgi compartments in *S. cerevisiae*	Neutral	Prohibitive	Not shown
Temporal fusion of heterogenous Golgi compartment in *S. cerevisiae*	Neutral	Desired	Shown

**Table 3 ijms-23-03590-t003:** Protocols of cargo synchronisation.

Protocol	State of Golgi	Steps of Cargo Synchronisation			
	Complex	Accumulation of Cargo in ER Exit Sites and Emptying of Golgi	Duration of Cargo Pulse	Block of Delivery of Visible Cargoes to the Golgi	Beginning of Video Recording
**1. Small pulse (40-15-40)**					
A. E-40-15-40-small (the mini-wave)	Empty	16 h at 40 °C	15 min at 15 °C	Shift back to 40 °C	2 min after shift to 40 °C
B. F-40-15-40-small (the mini-wave)	Full				10 min after shift to 40 °C
**2. Large pulse (40-15-40)**					
A. E-40-15-40-large (the maxi-wave)	Empty	16 h at 40 °C	2 h at 15 °C	Shift back to 40 °C	2 min after shift to 40 °C
B. F-40-15-40-large (the maxi-wave)	Full				10 min after shift to 40 °C
**3. Pulse (40-32-40)**					
A. E-40-32-40-small (the 40-32 mini-wave)	Empty	16 h at 40 °C	5 min at 32 °C	Shift back to 40 °C	2 min after shift to 40 °C
B. F-40-32-40-small	Full				10 min after shift to 40 °C
**4. Emptying–pulse (32[CHM]-15-32[CHM])**					
A. E-CHM-15-CHM	Empty	16 h at 40 °C, then CHM for 3 h at 32 °C	2 h at 15 °C (without CHM)	Re-addition of CHM, shift back to 32 °C	2 min after shift to 40 °C
B. F-CHM-15-CHM	Full				10 min after shift to 40 °C
**5. ER accumulation–chase (40-32)**					
A. E-40-32 (iFRAP)	Empty	16 h at 40 °C	None	Bleaching of whole cell less the Golgi area	5 min after shift to 32 °C
B. E-40-32 (piFRAP)			3 min		
C. F-40-32 (iFRAP)	Full		None		30 min after shift to 32 °C
D. F-40-32 (piFRAP)			3 min		
**6. Steady-state iFRAP**					
A. Steady-state iFRAP	Full	16 h at 40 °C	None	Bleaching of whole cell less the Golgi area	Immediately after bleaching
B. Steady-state piFRAP	Full		3 min		
**7. Pulse–outcoming wave**					
A. F-40-32-40 outcoming	Full	16 h at 40 °C	30 min at 32 °C	Shift to 40 °C for 30 min	0, 2, 4, 6, 8, 10, 12 min after shift

ER/IC, endoplasmic reticulum/intermediate compartment; iFRAP, inverse fluorescence recovery after photobleaching; piFRAP, post- inverse fluorescence recovery after photobleaching.

## Data Availability

Not applicable. This study did not generate new unique reagents.

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
