# Peer review of "Comparison of the Cisterna Maturation-Progression Model with the Kiss-and-Run Model of Intra-Golgi Transport: Role of Cisternal Pores and Cargo Domains†"

_ijms, 2022, doi:10.3390/ijms23073590_

Round 1
Reviewer 1 Report
In this study, Beznoussenko et al. use various synchronized ER-Golgi trafficking assays coupled to fluorescence and electron microscopy imaging, in order to address the question of Golgi maturation. The data is clearly high quality and I am impressed by the large number of high-quality immunogold labelling EM images; this is a very complicated technique.
However, I have a major problem with this manuscript, and that is that I don’t understand it. It seems to be unnecessarily complex, overly complicated, and incomprehensible. Most key concepts are not introduced at all, the rationale for the experimental work flow is not explained, and too little context is provided. I honestly don’t understand this manuscript, even though I am working in the closely related field of Golgi and membrane trafficking. The manuscript is probably understandable if you are familiar with the entire oeuvre of the authors, but not for me. The difficulty is that many of the central concepts, techniques, and analyses are not explained at all, but assumed to be general knowledge, even though these are by no means standard in the cell biology field. A similar concern holds true for the experiments and analyses, which are non-standard and should be properly explained. Because I don’t understand this manuscript, I cannot really judge its scientific merit, novelty and solidity. However, I would be will to review a completely rewritten manuscript, with all the concepts, models, experiments, analyses, etc introduced in a comprehensible fashion. Below a (non complete!) list of things that need to be introduced and explained:
What is meant by “surrounded with pores from the cisterna by pores” (line 17 -18).
What is meant with consumption and restoration of pores?
What are the 4 different models, why only focus on two models (CMPM and KARM)? What about the other two (VM and DM)?
Why does the KARM model have the perquisites listed in line 37-45?
Why is the concentration of mega-cargoes at the trans-side during IGT a prohibitive observation for the CMPM model (line 52-54). What is actually meant with mega-cargoes?
What is meant with ChMs (line 58)?
What are “cisternal rims” and PCI distentions?
What is steady state and synchronous IGT?
What are the empty zones in Figure 1, why are they important?
What cell types are used for what figures (numerous cell models listed in Methods section)?
Where are the mouse experiments, mentioned in the Methods section, used?
What systems are used for synchronization of traffic from which figures?
Why are SAGPR, procollagen I and VLDL selected as cargoes?
What is a “transport block” (line 82)?
What is the acronym PC in line 86?
What is CHM-15-CHM (line 119)?
What are the mini-wave and maxi-wave protocols?
How are the gold labelings quantified? How are they allocated to regions of the Golgi? In the EM, how is co-localization defined (for example in Fig. 4)?
How often were experiments analysed? Can you show individual datapoints and statistical P values? Can the authors show quantifications for the optical microscopy?
Minor comment:
The reference to figure 6 seems incorrect in the paragraph starting from line 244.
Author Response
Reviewer 1
In this study, Beznoussenko et al. use various synchronized ER-Golgi trafficking assays coupled to fluorescence and electron microscopy imaging, in order to address the question of Golgi maturation. The data is clearly high quality and I am impressed by the large number of high-quality immunogold labelling EM images; this is a very complicated technique.
Our reply. Thanks a lot for your kind words about the quality of images. We also appreciate your comments which for sure helped us to improve the quality of out work.
Reviewer 1. 1. However, I have a major problem with this manuscript, and that is that I don’t understand it. It seems to be unnecessarily complex, overly complicated, and incomprehensible. Most key concepts are not introduced at all, the rationale for the experimental work flow is not explained, and too little context is provided. I honestly don’t understand this manuscript, even though I am working in the closely related field of Golgi and membrane trafficking.
Our reply to 1.1. This issue of IJMS is a thematic issue devoted to highly specialized aspect of intracellular transport, namely, “Models of transport”. This issue is even more specialized than the Traffic journal where the reviewer proposed to send our paper. The issue of transport model is especially difficult for those who does not have the corresponding background.
Reviewer 1. 2. The manuscript is probably understandable if you are familiar with the entire oeuvre of the authors, but not for me.
Our reply to 1.2. Yes, this manuscript is for people who know these models of transport. For those who do not know the problem we quoted our papers where we described in details all these unclear questions. In this paper it is no possible to make this job du to limitation of the paper volume. For us it is very strange when the person not working in the Golgi filed agrees to be q[a reviewer in this highly specialized topic. Maybe this is a mistake of the editor who sent our paper not to the real specialist in this field. In any case, these comments are extremely useful for us and we are sure that after all necessary corrections the paper became more understandable for general readers. This is a huge work and we are very grateful to our reviewer for this job. We also thank our reviewer for his decision not to reject our manuscript (as so-called established scientists do in so-called top journals) but to ask major revision.
Reviewer 1. 3. The difficulty is that many of the central concepts, techniques, and analyses are not explained at all, but assumed to be general knowledge, even though these are by no means standard in the cell biology field. A similar concern holds true for the experiments and analyses, which are non-standard and should be properly explained. Because I don’t understand this manuscript, I cannot really judge its scientific merit, novelty and solidity. However, I would be will to review a completely rewritten manuscript, with all the concepts, models, experiments, analyses, etc introduced in a comprehensible fashion.
Our reply to 1.3. Thanks a lot. We improved the manuscript significantly and added a lot of schemes and movies describing these concepts.
Reviewer 1.3.A. Below a (non complete!) list of things that need to be introduced and explained: What is meant by “surrounded with pores from the cisterna by pores” (line 17 -18).
Our reply to 1.3.А. Thanks a lot. We improved the manuscript significantly and added a lot of schemes and movies describing these concepts.
Reviewer 1.3.B. What is meant with consumption and restoration of pores?
Our reply to 1.3.B. These words indicate that when a cargo domain pass across the Golgi stack it uses pores as the site containing thin tubules to make the fission of this domain easier. During this passage the number of [ores decreases and therefore we used the word consumption. Therefore, we used the word consumption because each cargo domain during this passage consume pores. However, in order to make the text simpler we eliminated these words and replaced them with the words “decrease” and “increase” of the number of pores.
Reviewer 1.3.C. What are the 4 different models, why only focus on two models (CMPM and KARM)? What about the other two (VM and DM)?
Our reply to 1.3.C. We described all models of intra-Golgi transport in our reviews (Mironov et al. 1997; 2005, 2013; 2017; Mironov and Beznoussenko, 2008, 2012; 2019; Beznoussenko and Mironov, 2002). Now there are more than 20 models explaining. There is no space to described all of them here. Similarly, there is not space and logic for the comparison of all these models altogether. Therefore, we decided to compare models using the pair-based approach. Here, we compared the cisterna maturation and KAR models. In the next paper ,we will compare the diffusion and the KAR model. After these comparisons and analysis of the models of ER-Golgi and Golgi-PM transport we plan to write the general paper.
Reviewer 1. 3.C. Why does the KARM model have the perquisites listed in line 37-45?
Out reply to 1.3.C: We described all this in. details in our reviews (Mironov and Beznoussenko, 2019). In the new version of the manuscript, we added the brief explanation of this question.
Reviewer 1.3.D. Why is the concentration of mega-cargoes at the trans-side during IGT a prohibitive observation for the CMPM model (line 52-54). What is actually meant with mega-cargoes?
Out reply to 1.3.C. We described all this in details in our reviews (Mironov and Beznoussenko, 2019). In the new version of the manuscript, we added the brief explanation of this question. The main postulate of the CMPM is that cargo never leave the lumen or membrane of just formed cis-Golgi cisterna during its progression across the stack. As such, there is not possibility to increase concentration of a cargo within the framework of this model. The word “Mega-cargo” indicates the cargo with the size higher than the diameter of COPI vesicle. Additionally, these cargos are subjected to disassembly during their travel across the Golgi stack. The size of mega-cargo prevents its transport by COPI-dependent vesicles.
Reviewer 1. 3.E. What is meant with ChMs (line 58)?
Our reply to1.3.E. ChM means chylomicron. We eliminated this abbreviation and used the work chylomicron everywhere.
Reviewer 1. 3.F. What are “cisternal rims” and “PCI distentions”?
“Cisternal rims” is the widely used term in the field of Golgi and indicates the edges of cisternal disks. “PCI-distension” indicates the local widening of the cisterna, lumen which is filled with the aggregate of procollagen triplexes.
Reviewer 1. 3.G. What is steady state and synchronous IGT?
Steady state indicates situation when cells are transported cargo under normal condition without any attempt to accumulate and then to release our cargo. Synchronization means the experimental action which leads to the blockage of the cargo exit from one of the secretory compartments and the cargo accumulation in the proximal compartment. Then the transport block is released and accumulated cargo begins to move along the secretory pathway and a large wave. The synchronization is helpful for the visualization of this passage of our wave along the already emptied of the secretory pathway. However, synchronization induces overloading of the system.
Reviewer 1.3.H. What are the empty zones in Figure 1, why are they important?
Our reply to 1.3.H. When albumin arrives at the Golgi through connections (Beznoussenko et al., 2014) or with membrane domains, it diffuse quickly and should fill the entire lumen of the Golgi including central parts of cisternae. However, this is not the case. There could be at least two hypotheses explaining this phenomenon. 1. The lumen of cisterna in the central part of cisternae could be too narrow for unlimited diffusion. 2. This part of cisternae could be filled with protein domains restricting free diffusion of albumin. This phenomenon could also help albumin to be transported quicker than under the conditions, when the whole cisternal lumen is filled with albumin. Similar restriction for the diffusion of G proteins of VSV into the central part of Golgi cisternae was described by Rothman (Nobel prize).
Reviewer 1. 3. B. What cell types are used for what figures (numerous cell models listed in Methods section)?
Our reply to 1.3.H. We added this information.
Reviewer 1. 3.I. Where are the mouse experiments, mentioned in the Methods section, used
Our reply to 1.3.I. This is a mistake. We used rats. We corrected this mistake.
Reviewer 1. 3.J. What systems are used for synchronization of traffic from which figures?
Our reply to 1.3.J. We added this information.
Reviewer 1. 3.K. Why are ASGPR, procollagen I and VLDL selected as cargoes?
Our reply to 1.3.K. The reasons are the following. 1. These cargos are conventional and extensively examined. Analysis of these cargoes is convenient because several tagged constructs exist and good antibodies are present. This aspect is extremely important. Also, in our stud we should use three main types of cargos, namely, soluble, albumin; membrane cargos (VSVG and ASGPR; in order to test whether membrane cargos are subjected to intermixing); procollagen as an example of a soluble cargo, which form stable aggregates inside cisterna lumens. We worked with these cargos for many years and know them rather well.
Reviewer 1.3.L. What is a “transport block” (line 82)?
Our reply to 1.3.L. This term is widely used for the description of the situation when using different methods, we block exit of cargo from some compartment. Transport of cargo is stopped and we used the term “transport block”.
Reviewer 1.3.M. What is the acronym PC in line 86?
Our reply to 1.3.M. “PC” indicates procollagen. We eliminated this abbreviation and replaced it everywhere with the word procollagen.
Reviewer 1.3.N. What is CHM-15-CHM (line 119)?
Our reply to 1.3.N. All protocols of the cargo synchronization were explained in Table S4. The CHM-15-CHM synchronization protocol is composed of the initial treatment of cells with cycloheximide at 37˚C in order to block synthesis of all cargos in the ER. Then in 1 h, CHM was eliminated and cells were placed at 15˚C for 15 min or 2 h. Next, cells were placed at 37˚C and CHM is added again. Under these conditions, only cargo which was concentrated within the ER exit sites would travel across the Golgi.
In the new version of our manuscript, we placed all supplementary material in the main text, because the journal is based on the computer storage of the papers but not on the paper based issues. There is no strong limitation in the length of the paper. We also introduced short description of the protocols in the text.
Reviewer 1.3.O. What are the mini-wave and maxi-wave protocols?
Our replay to 1.3.O. All these protocols were described in our previous Table S4. Now this Table 4 is the main text and additional brief descriptions were done in the “Material and Method” chapter and in the “Results”.
Reviewer 1.3.P. How are the gold labelings quantified? How are they allocated to regions of the Golgi? In the EM, how is co-localization defined (for example in Fig. 4)?
Our reply to 1.3.P. We presented all quantitative data and explained how we measured our results in Material and Methods. We estimated labelling densities according to Lucocq (1993) and Mayhew et al. (2002, 2008). Briefly, the length of membrane contours was measured using square test grid and is equal to the number of intersections multiplied by the length of one unit (depending on magnification and test line density) of this grid (see book by Griffith book) Then we countered the number of gold particles localized on this contour. The labelling density (LD) was equal to the gold number/length. In each dish from each pair, we measured two cells and estimated the mean for control and the mean for experiment. Then we normalized the value of the experimental dish as a percentage of control. These normalize values were used for the statistical evaluation of the mistake probability (P) where N=6. This number was estimated in our papers (Mironov and Mironov, 1998).
(Mayhew TM, Lucocq JM, Griffiths G (2002) Relative labelling index: a novel stereological approach to test for non-random immunogold labelling of organelles and membranes on transmission electron microscopy thin sections. J Microsc 205:153-164. Mayhew TM, Lucocq JM (2008) Quantifying immunogold labelling patterns of cellular compartments when they comprise mixtures of membranes (surface-occupying) and organelles (volume-occupying). Histochem Cell Biol 129:367-378. Mironov, A. A. Jr. and Mironov, A. A. 1998. Estimation of subcellular organelle volume from ultrathin sections through centrioles with a discretized version of vertical rotator. J. Microsc. 192:29-36.).
Reviewer 1. 3.Q. How often were experiments analysed? Can you show individual datapoints and statistical P values? Can the authors show quantifications for the optical microscopy?
Our reply: We presented all quantitative data and explained how we measured our results in Material and Methods. Briefly: In all experiments we used 6 pairs of samples composed of one control and one experimental dish (or rats). Each pair is treated under completely identical conditions. Even sectioning was organized in such a way when control and experimental samples were embedded in one section block. We added these individual normalized values in each graph.
Reviewer 1.4.A. Minor comment: The reference to figure 6 seems incorrect in the paragraph starting from line 244.
Our reply to 1.4.A. We corrected this mistake.
Reviewer 2 Report
Quality of presentation is so low that was not possible to follow the experiments. Therefore, it was not possible to jugde if conclussions are supported by data. Quality of pictures is high and they may contain important data, howerer by the presentation is not acceptable.
Introductiuon is definitely not sufficient. Models of intra-Golgi transport should be explained. Previous contribution of the authors to the field should be given briefly. If authors address the manuscript to Golgi specialists only, they should send the manuscript to specialized journal.
Abbreviations are not introduced in the text. For example L 28 GC, L72 VLDL. Too many abbreviatios, does not help the reader.
Chapter Results is not well organized. Chapter 2.1 should be labeled earier, before FIg.1 is cited.
In Results experiments are not described at all, only conclusions are given. Which cells were used, what was the design, what was studied?
The legends of Figures should contain information what is on the picture and bierfly how the experiment was performed, what conditions. Labeling of parts of figures should be consistent, the same font, the same color and on the same site of the image. Labeling of parts, such as D, E, K should be given in left upper corner of the image and upper left site outside the graph. Parts of Figures should be ordered from the left to right or form the top to the bottom, not mixed. Scale bar should be given on each micriscopic image. Appropriate space should be given between parts of the figure. Otherwise is a mess and is difficult to follow.
Possibly the last author can read again and correct the manuscript.
Author Response
Reviewer 2
Reviewer 2.1. Quality of presentation is so low that was not possible to follow the experiments. Therefore, it was not possible to jugde if conclussions are supported by data. Quality of pictures is high and they may contain important data, howerer by the presentation is not acceptable.
Our reply to 2.1. We re-wrote the manuscript and added all necessary information. We thank our reviewer 2 for very valuable suggestions.
Reviewer 2.2. Introductiuon is definitely not sufficient. Models of intra-Golgi transport should be explained. Previous contribution of the authors to the field should be given briefly. If authors address the manuscript to Golgi specialists only, they should send the manuscript to specialized journal.
Our reply to 2.2. We extended our Introduction and added our contribution.
Reviewer 2.3. Abbreviations are not introduced in the text. For example L 28 GC, L72 VLDL. Too many abbreviatios, does not help the reader.
Our reply to 2.3. We corrected these mistakes and decreased the number of abbreviations.
Reviewer 2.4. Chapter Results is not well organized. Chapter 2.1 should be labeled earier, before FIg.1 is cited.
Our reply to 2.4. We re-wrote this chapter according to the reviewer’s suggestions.
Reviewer 2.5. In Results experiments are not described at all, only conclusions are given. Which cells were used, what was the design, what was studied?
Our reply to 2.5. We re-wrote this chapter according to the reviewer’s suggestions.
Reviewer 2.6. The legends of Figures should contain information what is on the picture and bierfly how the experiment was performed, what conditions.
Our reply to 2.6. We re-wrote these legends according to the reviewer’s suggestions.
Reviewer 2. 6. Labeling of parts of figures should be consistent, the same font, the same color and on the same site of the image.
Our reply: Although in the author’s instructions there are no such strong demands we thank our reviewer 2 and re-made all our Figures according to these demands.
Reviewer 2. 7. Labeling of parts, such as D, E, K should be given in left upper corner of the image and upper left site outside the graph. Parts of Figures should be ordered from the left to right or form the top to the bottom, not mixed. Scale bar should be given on each micriscopic image. Appropriate space should be given between parts of the figure. Otherwise is a mess and is difficult to follow.
Our reply to 2.7. We re-made all our Figures according to these demands.
Reviewer 2. 8. Possibly the last author can read again and correct the manuscript.
Our reply to 2.8. We re-wrote the manuscript completely and now hope that it became much better.
Round 2
Reviewer 1 Report
The authors have extensively revised the manuscript, and it substantially improved. The new diagrams are very helpful. I have no further comments and support publication.
Author Response
Reply
There was a demand to correct our English.
Our reply
Englishman corrected all mistakes
Academic Editor Notes
- The intra-Golgi transport is important for the ER-Golgi transport and trafficking system.
In introduction, a major part of Glycan-based trafficking should be described in Introduction and Discussion.
Our reply
We agree with the scientific editor and added the following part in the text (Line 104-115):
“Here, we do not analyse mechanisms of IGT of glycan and mucins because this issue is extremely complicated. For instance, there are N-Linked glycans and O-Linked glycans, which behave as membrane glycosylated proteins. On the other hand, secreted mucins could be secreted (gel-forming [MUC2, MUC5AC, MUC5B, MUC6, MUC19] and non-gel-forming [MUC7, MUC8, MUC9, MUC20]); membrane-bound and transmembrane (MUC1, MUC3A, MUC3B, MUC4, MUC12, MUC13, MUC15, MUC16, MUC17, MUC21, MUC22). However, these proteins could be classified in related to IGT. Glycans, membrane-bound and transmembrane mucins behave as conventional glycosylated membrane cargo-proteins (i.e., VSVG). The gel-forming glycans correspond to the secretory cargo-aggregates (i.e., procollagen-I), non-gel-forming mucins mimic diffusional secretory cargoes (i.e., albumin [19,25]). Although the intra-Golgi transport is important for the ER-Golgi transport and trafficking system, we examined here only the first issue”.
Additionally, we corrected all mistakes in Images and legends.